# Multiple lines of evidence for a hypervelocity impact origin for the Silverpit Crater

Uisdean Nicholson [1] ✉, Iain de Jonge-Anderson [1,10], Alex Gillespie[2], Thomas Kenkmann [3], Tom Dunkley Jones [4], Gareth S. Collins [5], James Frankel[2], Veronica Bray [6], Sean P. S. Gulick [7,8] & Ronnie Parr[9]

An impact origin for Silverpit Crater, on the UK continental shelf, has been contested over the last two decades, with a lack of definitive evidence – traditionally petrographic evidence of shock metamorphism – to resolve the debate. Here we present 3D seismic, petrographic and biostratigraphic data, and numerical impact simulations to test the impact hypothesis. The seismic data provide exceptional imaging of the entire structure, confirming the presence of a central uplift, annular moat, damage zone and numerous secondary craters on the contemporaneous seabed. The distribution of normal and reverse faults in the brim, and curved radial faults around the central uplift suggest a low-angle impact from the west. The pitted, flat-topped central uplift at the top chalk horizon may indicate significant devolatilization of chalk immediately following impact. Biostratigraphic data confirm that this event occurred during the middle Eocene, between 43-46 million years ago. Petrographic analysis from the reworked ejecta sequence in the nearby 43/25-1 well reveals two grains with shock lamellae, indicating shock pressures of ~10-13 GPa, consistent with results from our numerical models. This combination of data and modelling provide compelling evidence that Silverpit Crater is an exceptionally preserved hypervelocity impact structure.

Hypervelocity impacts of asteroids and comets with Earth represent a significant hazard and are a ubiquitous process in the Solar System. Impactors larger than ~100 m in diameter are capable of penetrating the atmosphere and striking Earth's surface to form craters[1]. Such events are rare, with none observed in recorded history; therefore, their consequences can only be inferred from impact craters and ejecta deposits preserved in the geological record. Impact structures are relatively rare on Earth, with only ~200 confirmed impact craters in the terrestrial record[2,3]. Marine-impact craters are even more rarely preserved, despite over 70% of the Earth's surface being covered with water, with only ~33 confirmed or probable marine-impact craters identified[4].

Most terrestrial impact structures are poorly preserved as those exposed at the Earth's surface are modified by weathering, erosion and tectonic deformation. Buried craters are often better preserved, but these are difficult to investigate without high-resolution seismic imaging and/or drill cores. The only impact crater on Earth that is fully imaged with 3D seismic is the ~9 km diameter Nadir Crater offshore

[1]School of Energy, Geoscience, Infrastructure and Society, Heriot-Watt University, Edinburgh, UK. [2]Northern Endurance Partnership (NEP), bp, Sunbury-on-Thames, UK. [3]Institut für Geo und Umweltnaturwissenschaften, Universität Freiburg, Freiburg im Breisgau, Germany. [4]School of Geography, Earth and Environmental Science, University of Birmingham, Birmingham, UK. [5]Department of Earth Science & Engineering, Imperial College London, London, UK. [6]Lunar and Planetary Laboratory, University of Arizona, Tucson, Arizona, AZ, USA. [7]Center for Planetary Systems Habitability, University of Texas at Austin, Austin, TX, USA. [8]Institute for Geophysics & Department of Earth and Planetary Sciences, University of Texas at Austin, Austin, TX, USA. [9]North Sea Transition Authority, Aberdeen, UK. [10]Present address: Department of Civil & Environmental Engineering, University of Strathclyde, Glasgow, UK. ✉e-mail: u.nicholson@hw.ac.uk

West Africa[5,6]. Other craters identified on 2D seismic have either been deformed by later tectonic processes (e.g. Mjolnir[7]) or are too large to be imaged by 3D seismic datasets (e.g. Chesapeake[8], Chicxulub[9]). This limits our understanding of the near-surface processes that occur during and shortly after an impact event, which likely vary depending on the environment of impact and the physical properties of the target stratigraphy.

Traditionally, proposed craters are only confirmed as hypervelocity impact structures on the basis of shocked mineral phases—principally quartz or feldspar—and associated evidence from physical samples[10,11]. Shocked minerals only form at extreme transient shock pressures that cannot be replicated by other terrestrial processes, even deep within the lithosphere. The Nadir Crater is the only example where seismic data alone has been used to show 'beyond a reasonable doubt' an impact origin[6], showing that there are exceptional cases in which high-quality geophysical imaging allows near-diagnostic structural characteristics to be identified and other epigenetic mechanisms to be conclusively ruled out[6].

The Silverpit Crater is situated in the Silverpit Basin in the southern North Sea, UK (Fig. 1). The basin has a complex tectonic history, having experienced multiple periods of rifting and uplift throughout the late Palaeozoic, Mesozoic and Cenozoic[12,13]. At the Silverpit Crater location, Triassic and Jurassic evaporites, carbonates and clastic sedimentary rocks overlie a mobile Zechstein evaporite sequence (Fig. 2). The Jurassic and Triassic sediments are partially eroded across the area by the Base Cretaceous Unconformity (BCU), which is overlain by Cretaceous limestones and chalk, and Paleogene marine mudstones. The Triassic to Paleogene section has been folded into a series of gentle 10–100 km scale, NW-SE trending anticlines and synclines, which have been subsequently eroded and then buried during the Quaternary[14].

The crater was identified from 3D seismic data in 2002, and proposed as a new candidate hypervelocity impact structure[15]. Although the seismic data available at the time only covered part of the

structure, the crater was described as a 20-km-diameter, multi-ringed structure with characteristics consistent with a complex impact crater, including a circular planform morphology, concentric faults, and a proposed stratigraphic uplift below the crater floor. The crater floor was inferred to be at or just above the Cretaceous-Paleogene (K-Pg) boundary, with an age of approximately 65-60 Ma, based on extensive deformation of the Upper Cretaceous chalk below the crater floor. More seismic data were acquired and interpreted in the following years, further constraining the geometry of the potential impact structure[16]. Although the structure was still only partially imaged, these data allowed the crater rim to be redefined as a much smaller ~8 km in diameter, based on the most distant inward-facing extensional fault defining a fault terrace at the top of the Cretaceous. According to this model, concentric extensional faults beyond the rim were interpreted to have formed over longer timescales (thousands of years) due to viscous relaxation of the sediments near the seabed into the crater cavity. The stratigraphic position of the crater floor was later interpreted to be shallower than previously thought, based on the presence of faults (albeit poorly imaged because of seismic multiples) cutting across the lower Paleogene interval. This led to the proposition of a much younger, Eocene, impact age[17]. This age was interpreted based on the correlation of seismic with unpublished biostratigraphic data from industry well 43/25-1, which penetrates the stratigraphy around 3 km to the NW of the centre of the proposed rim.

An impact origin for the crater has been disputed, with crater formation instead being explained by salt withdrawal in the deep subsurface[18], or by hydrothermal venting associated with Paleogene dykes that are present across the wider Silverpit Basin[19]. Some of the structural evidence for an impact origin is also disputed, with the stratigraphic uplift inferred to be a seismic artefact at the boundary of several seismic surveys, where full-fold seismic coverage was absent[18]. These competing hypotheses were widely reported in the media, resulting in a public debate at the Geological Society of London in 2009, where those present voted "overwhelmingly" for a non-impact origin (https://www.geolsoc.org.uk/Geoscientist/Archive/December-2009/Silverpit-not-crater). Even though this was a public vote rather than a decision arrived at by an expert panel of impact specialists, this debate's outcome, combined with a lack of new evidence, appears to have led many researchers to consider the question closed, with limited further research on subsurface data or modelling in the decade and a half since.

Here, we reassess the impact hypothesis for Silverpit Crater with geophysical data, analysis of well cuttings, and numerical modelling. We use high-resolution, pre-stack depth migrated (PSDM) 3D seismic data acquired by Northern Endurance Partnership (NEP) that offers improved imaging of the crater, and full seismic coverage of the central uplift. Samples from drill cuttings (rock fragments transported back to the surface in drilling mud) from well 43/25-1 are analysed to constrain the age of the crater biostratigraphically, and to look for evidence of shock metamorphism in the proposed proximal ejecta deposits, or in the subsurface at the time of impact. Finally, we present numerical impact simulation results to estimate the impact energy required to produce the observed crater morphology in the submarine, sedimentary target, and particularly to test models of crater modification based on observational evidence. In combination, these data show strong evidence for a hypervelocity impact origin for this structure.

## Results
### Seismic observations
The Silverpit Crater is here defined as a 3.2 km-diameter circular depression with its centre situated around 3 km to the southeast of the 43/25-1 well (Figs. 2 and 3). The surface is characterised by two distinct seismic reflections, herein referred to as CF1 and CF2, separated by a seismically transparent crater-fill package, sitting within the Paleogene

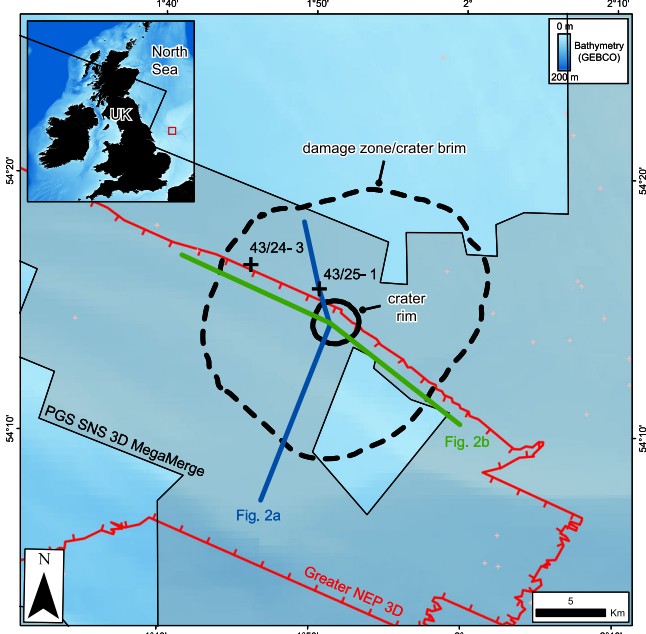

**Fig. 1 | Location map showing the Silverpit Crater and its associated damage zone.** Also shown are the locations of key wells (43/25-1 and 43/24-3), the outlines of 3D seismic datasets including the Greater NEP 3D (red outline) and the PGS SNS 3D MegaMerge volumes (shaded) and the locations of regional cross-sections presented in this study (Fig. 2). The inset map shows the location of the main map in the small red square. Bathymetry is from GEBCO[67].

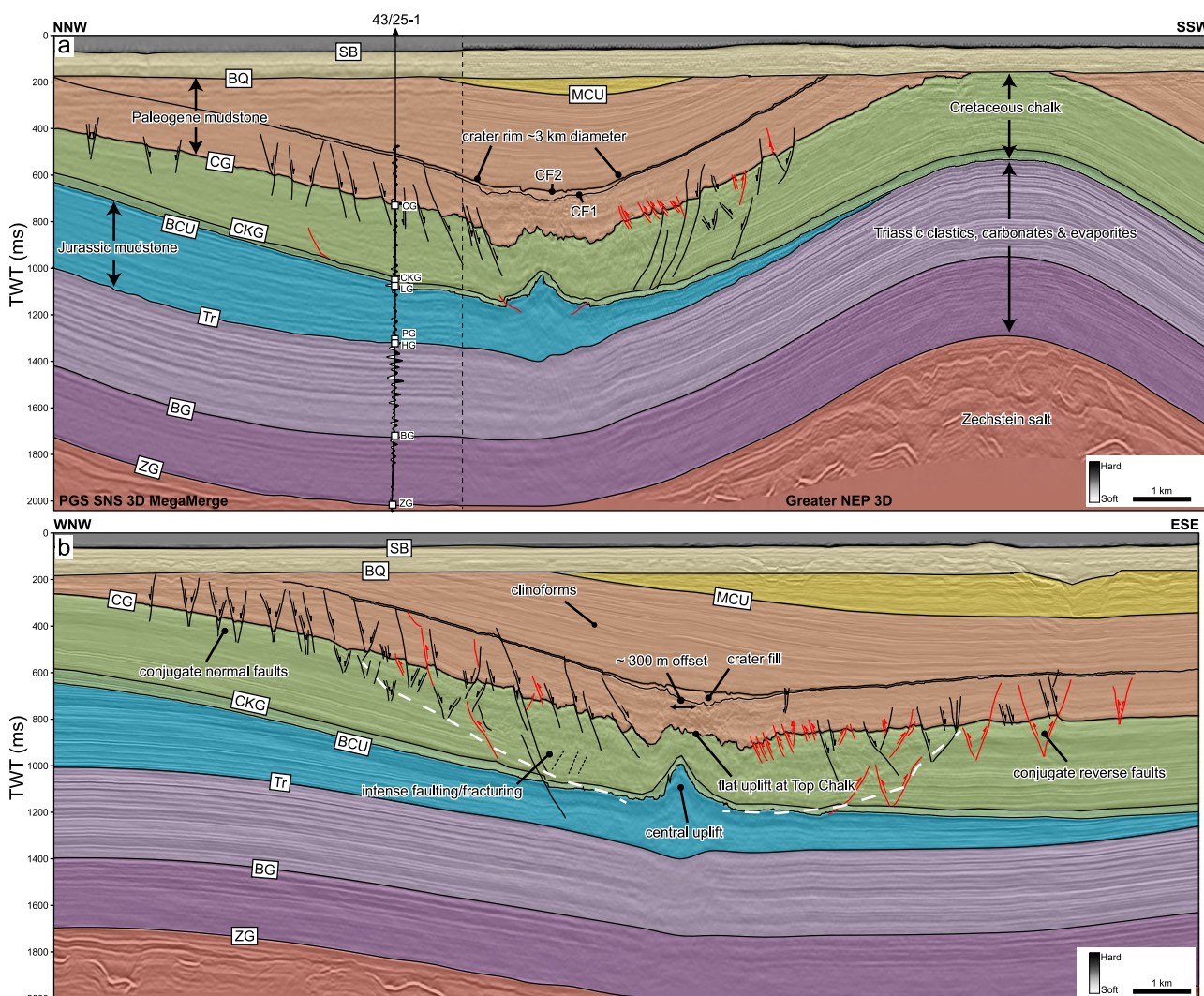

**Fig. 2 | Regional seismic profiles across Silverpit Crater, showing crater morphology and deformation patterns. a** is oriented NNW-SSW, intersecting the 43/25-1 well, and **b** is oriented WNW-ESE, approximately parallel to the inferred impact trajectory for the crater (see Fig. 1 for line locations). Note that conjugate faults are predominantly normal (black) uprange, and predominantly reverse faults (red) downrange, assuming an impact from the west. The 3.2 km diameter crater at mid-section. Eocene level is offset by around 0.3 km downrange relative to the central uplift at BCU level. See Supplementary Fig. S1 for an uninterpreted version of these lines and S2 and S3 for details about the seismic-to-well tie in (**a**). The white dashed line in (**b**) shows the extent of the more intense damage zone in the chalk, with multiple smaller faults at or below seismic resolution.

section. These reflections are located above a zone of broad deformation which deepens and intensifies towards the crater centre. Below the margins (rim) of the crater, the chalk in particular is deformed by normal faults into an annular moat, surrounding an uplifted, anticlinal structure below the centre of the crater at both top chalk and BCU level.

The Paleogene crater floor and contemporaneous seabed shows a stepped morphology defining three concentric circular zones: a ~1.2 km diameter nested central crater, a ~3.2 km diameter crater rim corresponding to a subtle topographic peak (*sensu*[20]) (Figs. 3 and 4), and a ~18 km diameter zone coincident with the extent of underlying deformation, referred to here as the crater brim (*sensu*[21]) (Fig. 5). The thickness of the crater-fill package mirrors this pattern with ~60 m, ~30 m and <10 m present within the nested inner crater, crater rim and brim respectively (Fig. 4). The latter is within tuning thickness (vertical resolution) of the seismic data and may be thinner. The top of the package, CF2, is characterised by a strong "hard" reflection (downward increase in acoustic impedance, caused by an increase in velocity and/or density) and the base, CF1, by a "soft" reflection, showing that the crater-fill has a higher acoustic impedance than the overlying and underlying lithologies.

The crater rim diameter of 3.2 km defined here is smaller than that described previously[16], although it is consistent with the interpreted scale of the original transient crater by these authors. This implies that the crater size is just above the simple-to-complex transition diameter of 2 km for sedimentary crater targets on Earth[22]. Previous larger crater size estimates were based on a probable near-end-Cretaceous crater age, and geomorphological observations of the top Chalk Group. Although we do see some subtle radial structures at the seabed beyond our interpreted crater rim, these are related to deformation external to the crater; that is, the 'outer limit of deformation', as described below[20]. The crater depth (considered here equivalent to crater-fill thickness) to crater rim diameter (3.2-km) ratio of ~1:50, and the presence of a central uplift, is consistent with the suppressed morphology of marine and sedimentary target craters of this size that have undergone substantial gravity-drive collapse[6,22].

Both the CF1 and CF2 crater floor reflections have anomalously high amplitudes relative to the background Paleogene reflectivity (Fig. 3b, e). These high amplitudes extend ~1.5 km into the crater brim, where they gradually decrease due to tuning effects and/or a decrease in acoustic impedance contrasts. Seismic amplitude and dip attribute

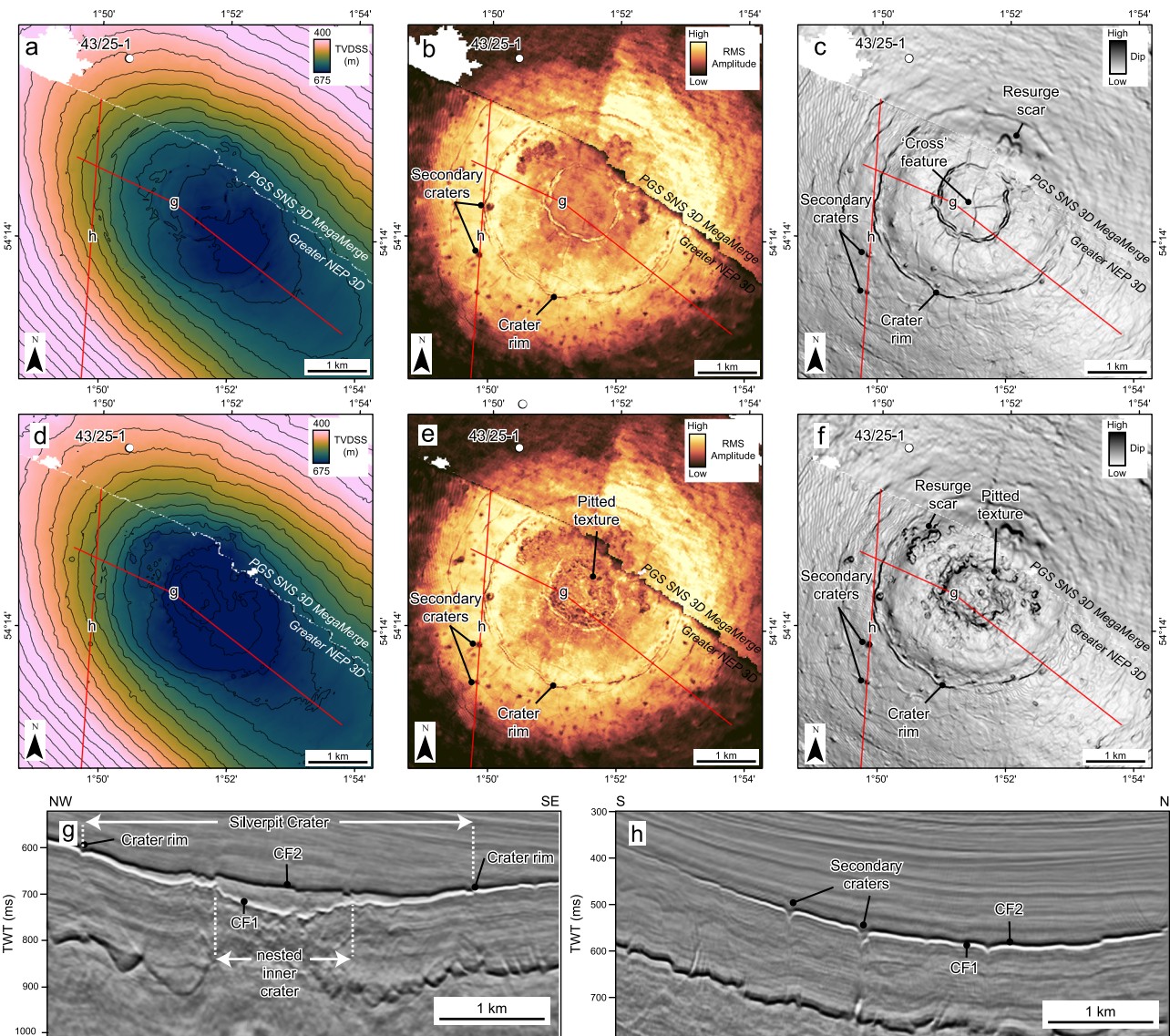

**Fig. 3 | Crater surface morphology and seismic attributes of horizons CF1 and CF2 at the crater floor.** True vertical depth subsea (TVDSS) structure maps with a 25 m contour interval (**a**, **d**), RMS amplitude (**b**, **e**) and dip (**c**, **f**) attributes for the CF1 (upper row) and CF2 (middle row) horizons, respectively. Seismic sections show the detailed crater morphology across the ~3.2 km central crater, including the ~1.2 km wide nested inner crater above the central uplift (**g**) and ~50–150 m diameter secondary craters beyond the crater rim (**h**). Note the high seismic amplitudes extending around 1.5 km from the crater rim in (**b**) and (**e**).

maps of the top and base crater-fill show several other distinctive features. CF2 displays two linear features within the inner crater, in a N-S and E-W orientation, respectively, producing a cross-like morphology (Fig. 3c). We interpret this to be a conjugate set of faults that form across the central crater floor after the crater was filled, presumably because of differential compaction. There are also several small ( ~ 0.25 ×1 km) scarps situated inside, but close to the crater rim (Fig. 3c, f), that we interpret as resurge scars formed as water cascaded back into the evacuated crater (e.g. refs. 6,23,24). These have not been observed in previously published geophysical data from Silverpit (c.f. [17]).

Between the crater rim and the outer extent of the high-amplitude fringe (~1.5 km from the rim) at the top and base crater surfaces, there are a series of individual circular depressions of up to ~150 m diameter (Fig. 3). Seismic sections across these circular features show that they are around 40 ms (~35 m) deep at the base crater reflection, but only up to 10 ms (~9 m) deep at the top crater reflection. We interpret these as secondary craters. These pits are up to 5% of the size of the primary crater (3.2 km) at Silverpit, and are thus consistent with scaling relationships for secondary craters observed on other planets and satellites[25]. An alternative explanation is that these could represent degassing features around the primary crater, which is discussed further below.

Immediately below the seabed contemporaneous to the crater floor is a ~18-km diameter zone of deformation which deepens and intensifies towards the centre of the crater. In the outer parts below the crater brim, concentric faulting is limited to the lower Paleogene to uppermost Cretaceous section (Figs. 2 and 3). Towards the centre of the crater, faulting extends increasingly deeper to BCU and Lower Jurassic levels. No faults are observed extending down into the Triassic section. A few faults extend across and slightly above the crater-fill package, where their displacement rapidly dies out. Extension of some structures above the crater-fill is interpreted as reactivation of crater-related faults during subsequent burial and folding.

The subsurface deformation is most clearly imaged at the top of the Upper Cretaceous chalk, where three distinct sets of faults can be interpreted (Fig. 5). The dominant faults are concentric, with mainly normal faults to the north, west and south, but mainly reverse faults in

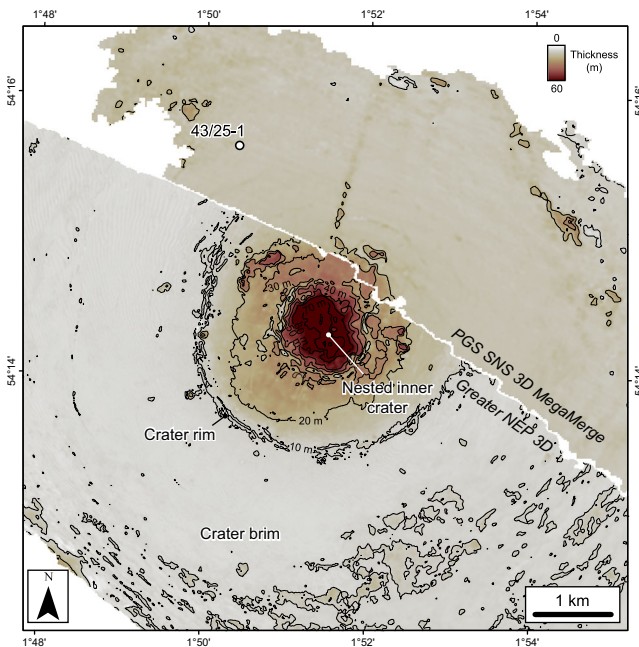

**Fig. 4 | Crater thickness map between crater floor horizons CF1 and CF2.** Contour increment is 10 m. The map shows a clear change in thickness between the central, nested inner crater (~60 m), the wider crater up to the crater rim (~15–30 m) and the crater brim (<10 m). The lack of contours in the PGS SNS 3D MegaMerge data highlights the increased frequency and lower tuning thickness of the Greater NEP 3D data.

the east (Fig. 2). Both normal faults and reverse faults form conjugate fault pairs, suggesting several décollement surfaces within the chalk, with a typical ~1 km spacing between fault sets. The improved image quality in the NEP seismic data shows unambiguously that most of these concentric faults extend all the way up through the Paleogene, to the crater floor horizon (cf. [15]). The contrast in deformation style is likely due to the rheological contrasts between different rock types, with the chalk deforming in a more brittle manner relative to the clay-rich Paleogene sequence above.

Cutting obliquely across the concentric faults are a set of curved, generally N-S trending faults that have a concave-to-the-east morphology (Fig. 5b). These have been previously described as 'spiral faults' and interpreted as having formed by centripetal radial extension following impact[16]. Our data show that these form exclusively on the north and east side of the crater, suggesting an asymmetric (oblique) impact. The sense of displacement on these faults varies and is difficult to fully constrain as these faults interact with the concentric faults. However, these have a strike-slip component and a reverse slip component in some locations (Fig. 5c).

In the central area below the crater, the deformation style at the top of the chalk changes. The entire Chalk Group shows intense faulting and fracturing below the crater rim, with deformation so intense that no consistent fault trends can be distinguished in seismic (Fig. 2). The Chalk Group also gets progressively thinner, towards the crater floor, from about 500 m in thickness below the outer brim, to ~250 m thickness below the crater rim (Fig. 6b). Correspondingly, the top chalk reflection deepens to form a ~1 km wide moat around a ~1 km wide, central plateau (Fig. 6a), which is elevated with respect to the moat but not relative to the undeformed chalk below the outer brim. Seismic attributes across both the moat and central plateau show a pitted texture, with individual pits of up to ~0.5 km diameter, that we interpret to be caused by fluid escape and/or devolatilization (Fig. 6b).

At the BCU level, there is a marked decrease in the intensity of deformation relative to the more brittle overlying chalk. At this depth,

faulting is limited to within a 2.5 km diameter zone below the crater. The main structural feature at the BCU level is a ~200 m high stratigraphic uplift, which is associated with a set of predominantly NE-SW orientated reverse faults radiating away from the uplift (Fig. 7). These radial faults have a slightly curved nature with a concave-to-the-west geometry and are consistent with those observed at other oblique complex impact craters[6,26–28]. The central uplift is offset from the centre of the Paleogene crater by around 300 m (Fig. 2b). The ratio of crater diameter to stratigraphic uplift at this depth is around 1:15.

Below the BCU uplift, the seismic data quality degrades, and reflections show a consistent push-down, suggesting that seismic velocities across this zone are lower than in the surrounding area. There is no evidence of a stratigraphic uplift below around 500–700 m beneath the crater floor.

## Nannofossil biostratigraphy

The 43/25-1 well is located 1 km to the northwest of the Silverpit Crater rim. It penetrates the inner part of the crater brim where seismic amplitudes at the CF1 and CF2 crater floor (and contemporaneous seabed) reflections are still elevated above background values (Fig. 3). The well-to-seismic tie (Fig. 8 and S2, S3) shows that the CF1 and CF2 reflections intersect the well at around 502 and 522 ms respectively (~471 m and ~495 m measured depth, MD). 31 samples were analysed for nannofossil assemblages using standard smear slides and polarising microscopy, between 183-607 m MD of well 43/25-1 to constrain the age of the crater floor (Fig. 8). The main results are described here and are presented fully in supplementary information.

All but one of the 23 samples taken between 439 and 597 m have a well-preserved, diverse but not abundant (5-10% of grains) typical middle-Eocene nannofossil assemblage. The age range of these sediments is constrained to be between 45.95 (Base *Sphenolithus furcato-lithoides*) and 43.06 Ma (Base common *Reticulofenestra umbilicus*), equivalent to Zones CNE10-12. The one smear slide barren of nannofossils in this interval was taken from the sample at depth 494 m, with a smear slide dominated by high-birefringence particles of ~2 to 15 μm diameter, in cross-polarised light (XPL) (Fig. S4) interpreted to be carbonate. In plain polarised light (PPL), this smear slide differs from the surrounding middle-Eocene sediments in having little organic matter or clay particles, and substantially less micron-scale pyrite, which is pervasive in the background of other middle-Eocene samples (Fig. S4). To test the consistency of observations from this depth, three more small (~0.1 g) cutting samples were used to make three additional smear slides. All three of these yielded middle-Eocene nannofossil assemblages similar to those samples above and below this level (Fig. S4).

Based on the well-to-seismic tie, the top crater floor reflections intersect the well at ~475-492 m. This position sits within the CNE10-12 nannofossil zone, showing that the age of the crater is between 43–46 Ma. The anomalous, barren sample at 494 m is near the base of this interval. The presence of a diverse middle-Eocene nannofossil assemblage in samples above and below the crater floor horizon provide clear evidence for an open marine setting, although these samples cannot be used to constrain the palaeo-water depth.

## Petrography and shocked minerals

Samples from the 421–610 m depth interval were investigated every ~6 m and analysed for petrography and potential deformation microstructures. Rock samples in the Paleogene are predominantly shales, with varying amounts of quartz, feldspars and clay grains. Petrographic results for these samples are summarised in Supplementary Information. We focus here on evidence for shock metamorphism in these samples, found exclusively in the 463–494 m interval, in the section equivalent to, and immediately stratigraphically above, the Silverpit crater floor (~492 m).

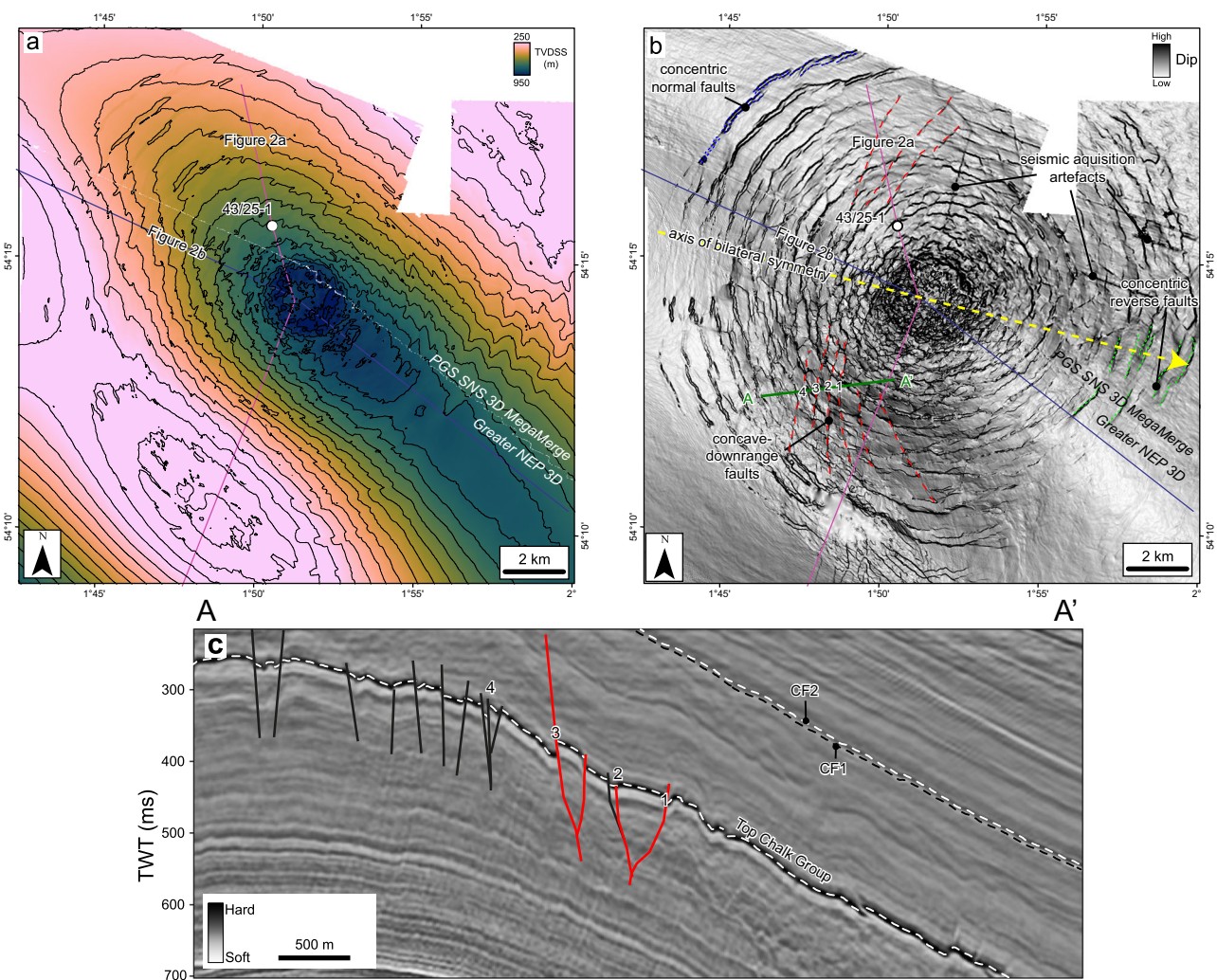

**Fig. 5 | Silverpit Crater deformation pattern at Top Chalk Group. a** True vertical depth subsea (TVDSS) structure maps generated from both the Greater NEP 3D and PGS SNS 3D MegaMerge seismic volumes. **b** Dip attribute map generated from the same horizon. Note concentric faults, with predominantly normal fault displacement to the north, west and south (example highlighted in blue), and predominantly reverse faults to the east (examples highlighted in green). Concave-to-the-east tangential faults (highlighted in red) are only found to the west of the crater (uprange direction). The yellow dashed line and arrow indicate the impact trajectory inferred from fault patterns. **c** Seismic profile across the concave-downrange faults, showing reverse movement on some faults (red). The line location and individual faults are shown on (**b**).

We found two grains of 40 to 80 μm size at 463 and 494 m depth that contain straight lamellae (Figs. 8 and 9). These lamellae are decorated with fluid inclusions, are strictly parallel, and have a spacing that ranges between 1 and 7 μm. One of the host grains is quartz (463 m), and one is potassium feldspar (k-felspar) (494 m). The quartz grain at 463 m shows at least two different crystallographic orientations of the lamellae along {10-13} and {10-14} in different domains of the crystal, which might indicate that the grain shows Dauphine-type twinning. Lamellae orientations are consistent with shock pressures of 10–13 GPa[10]. A third orientation could not be indexed. The k-feldspar grain at 494 m displays parallel lamellae decorated with fluid inclusions, and the lamellae have a slightly lower bulk density suggesting amorphization, consistent with shock metamorphism[29]. They are neither perthitic exsolution phenomena nor cleavage planes. The lamellae fulfil the characteristics of shock lamellae and are interpreted as decorated Planar Deformation Features (PDFs).

The presence of such shocked minerals at the stratigraphic equivalent of the crater floor provides strong independent evidence for an impact origin for Silverpit Crater. The presence of shocked minerals above the crater floor is discussed further below.

## Impact modelling

Reconnaissance simulations of the Silverpit impact were performed using a simplified target structure and vertical trajectory to constrain impactor parameters and timescales of different phases of deformation (Fig. 10). A five-layer target was used to represent the major rheologic divisions, including a nominal 100-m water layer, 300-m layer of weak, ductile Paleogene clay sediments, 600-m of brittle chalk, 300-m layer of ductile mudstone and underlying Triassic sediments (Fig. 10A). The simulation that produced the best match to the morphology and subsurface structure of the inner basin used an impactor diameter, speed and density of 160 m, 15 km s⁻¹ and 3300 kg m⁻³, respectively. An impact speed of 15 km s⁻¹, which is slightly lower than the average impact speed on Earth of 20 km s⁻¹, was selected to account for atmospheric deceleration. The simulation replicates the major structural observations of the inner basin observed in the seismic data, including the approximate morphology of the Top Chalk; the central uplift in the Jurassic mudstone and the envelope zone of pervasive damage (lighter shading in chalk layer). Concentric faults are not replicated in the wider brim, especially in the Paleocene, likely because of the oversimplification of the material model used to

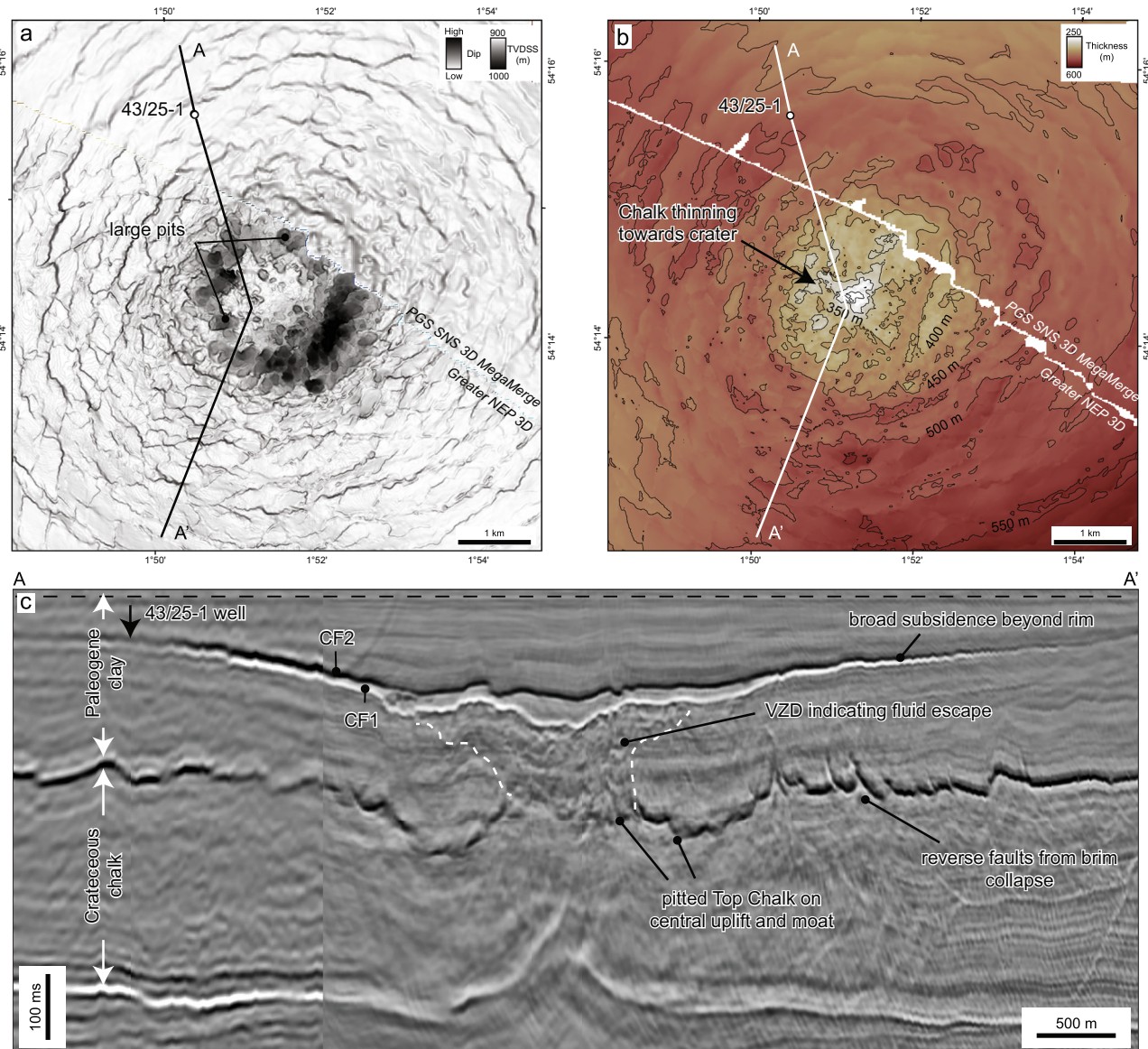

**Fig. 6 | Chalk deformation structures below the crater floor and brim, showing extensive surface pits and volume loss following impact. a** True vertical depth subsea (TVDSS) structure maps blended with the Variance attribute for the Top Chalk Group, generated from both the Greater NEP 3D and PGS SNS 3D MegaMerge seismic volumes. The central uplift and moat display pervasive surface pits of 50-500 m diameter. These may indicate significant devolatilization of the chalk following impact, with vaporised water and $CO_2$ from carbonate catastrophically released below the crater centre during the crater modification stage. **b** Chalk thickness map, showing substantial thinning of the chalk below the crater floor. Contour increment is 50 m. An estimated rock volume of 0.9–2.2 km$^3$ is missing from the chalk below the crater floor. **c** Seismic section flattened at a shallow horizon (black dashed line) within the Paleogene clay, to remove the effect of post-impact folding and reconstruct crater geometry. The pitted Top Chalk sits below a vertical zone of disruption (VZD) that we interpret as evidence of fluid escape (devolatilization) during the crater modification stage.

represent the weak, water-saturated, and partially lithified sediments in the Paleocene and chalk (cf. [5]).

The impact simulation forms a 1-km deep, 3-km wide transient crater within 12 s, which is lined with highly shocked clay and chalk sediments (Fig. 10B, C). Shock pressures in the chalk central uplift range from 1 GPa at a radius of 1 km to 20–30 GPa or higher near the top of the chalk and inside a 500-m radius. Particle tracking shows that the ejecta, with a range of shock pressures up to ~30 GPa, is derived entirely from the Paleogene clay, which is almost entirely excavated above the chalk inside the crater. Collapse of the crater, which enlarges the transient crater diameter by >30%, begins with uplift of the weaker mudstone and overlying chalk (30 s, Fig. 10D), which is subsequently covered by inward collapse of the overlying weaker and more mobile clay (60 s, Fig. 10E). An extended period (minutes) of resurge then

floods the crater and erodes the rim (Fig. 10F). At the location of the 43/25-1 well (~2.5-km radius), proximal ejecta, with a range of shock pressures, lands after about 30 s, settling through the water column as a density current and generating a large amplitude, breaking rim-wave tsunami.

## Discussion

### Evidence for an impact origin for Silverpit Crater

The combination of 3D seismic observations, numerical modelling results, and the presence of shocked minerals adjacent to the crater floor provides compelling evidence for a middle-Eocene (43–46 Ma) impact origin for Silverpit Crater.

The Greater NEP 3D seismic data provide strong independent evidence for a hypervelocity impact origin for the Silverpit Crater, even

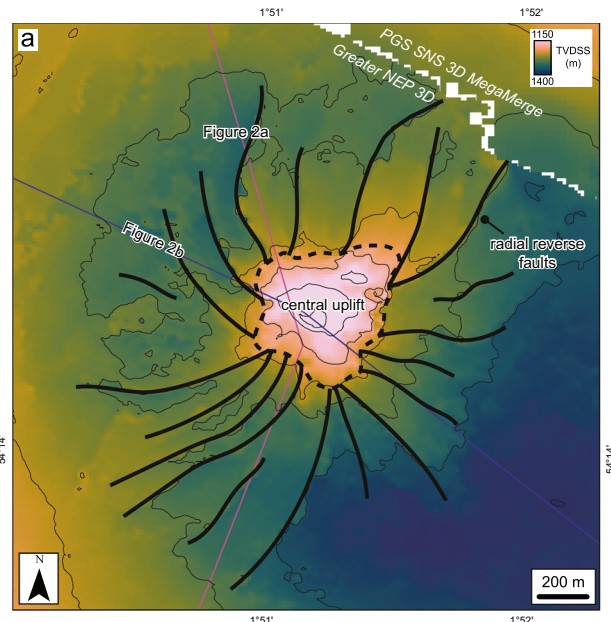
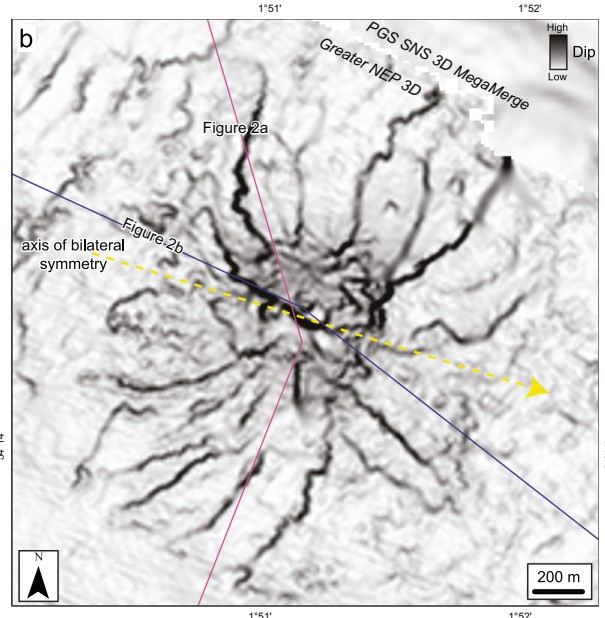

**Fig. 7 | Radial faults around the central uplift at the BCU level allow reconstruction of the impact trajectory.** True vertical depth subsea (TVDSS) structure maps (**a**) and dip attribute maps (**b**) of the Base Cretaceous unconformity, generated from both the Greater NEP 3D and PGS SNS 3D MegaMerge seismic volumes. The yellow arrow shows the axis of bilateral symmetry representing the impact trajectory.

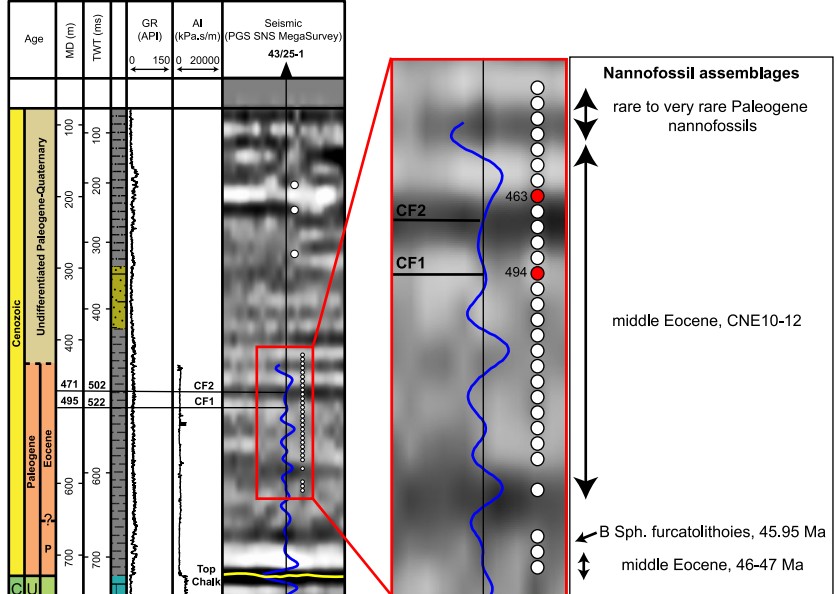

**Fig. 8 | Sample locations, nannofossil assemblages and shocked minerals from the 43/25-1 well.** Well-to-seismic tie of the Paleogene interval shown on the left with the PGS SNS 3D MegaMerge seismic data. The full tie for the entire well is shown in Figure S2 and smear slides are shown in Figure S4. Samples analysed for petrography and nannofossils are shown by white circles across the crater floor sequence. The red circles at 463 and 494 m show the samples with shocked minerals (Fig. 9). These are both within the middle-Eocene CNE10-12 sequence (43–46 Ma) and sit within and immediately above the crater floor sequence.

without petrographic evidence. These data provide full-fold seismic coverage of the entire crater, including the stratigraphic uplift below the crater floor. The data show that this uplift is undoubtedly a real geological feature, and not a seismic artefact, as suggested previously[18]. The ratio of the 3.2 km crater diameter to ~200 m stratigraphic uplift (1:16), ratio of crater depth ( ~60 m maximum at CF2) to diameter (1:50), and the depth of the stratigraphic uplift, extending to around 500–700 m below the crater floor, are all consistent with a terrestrial impact crater of this size[22]. We note that the crater diameter

that we document is considerably smaller than that reported previously[15,16]. Based on analogues and modelling results (Fig. 10), the multi-ring structure is interpreted to have formed by collapse of weak, unconsolidated sediment into the crater during the late crater modification stage[6], with the outer ring corresponding to the 'outer limit of deformation', well beyond the crater rim (*sensu*[20]). These concentric faults are associated with a subtle depression at the seabed, sometimes referred to as the crater brim, characteristic of soft-sediment or marine-target craters[21]. Such structures have similar characteristics to

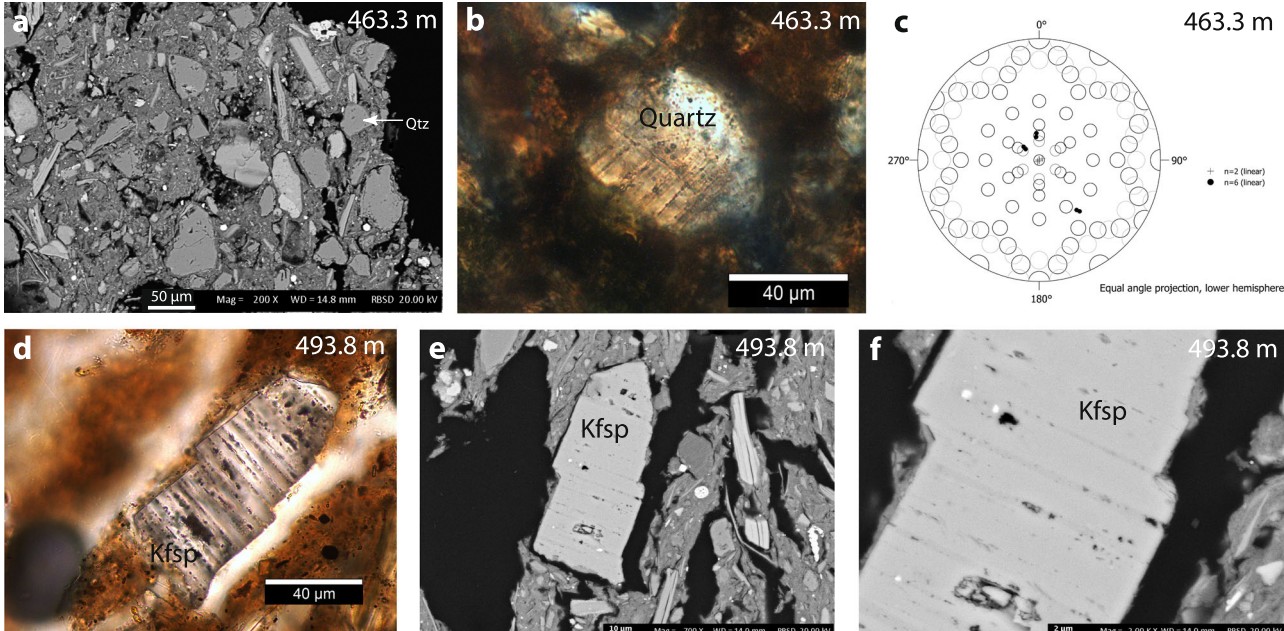

**Fig. 9 | Shock lamellae in grains in middle-Eocene sediments from the 43/25-1 well. a** SEM image of cuttings sample 463 m, showing larger silt grains in a clay-rich matrix. Labelled quartz grain is shown in (**b**). **b** Quartz grain from 463 m showing shock lamellae. **c** U-stage results for quartz mineral in b, showing straight lamellae with a {10-13} and a {10-14} orientation. **d–f** Transmitted-light photo-micrograph (**d**) and SEM images (**e**, **f**) of a k-feldspar grain from 494 m, showing amorphous lamellae decorated with fluid inclusions. Note that the lamellae behaved in a cohesive manner during deformation.

multi-ringed structures on icy satellites, such as Tyre or Callanish on Europa, although the rheological contrasts in these cases are different (soft-sediment over brittle chalk for Silverpit; ice over liquid water for icy satellites).

The lack of structural deformation deeper than ~700 m below the crater floor also shows that this is not a feature that would be sourced from below, such as a salt diapir or volcanic feature. Igneous dykes elsewhere in the Silverpit Basin are shown in some cases to be associated with linear arrays of pit chain craters of up to 2 km diameter, and Silverpit has previously been proposed to form by a similar mechanism[19]. However, the high-resolution NEP seismic data show no evidence of volcanic dykes in the vicinity of the crater. Moreover, our age constraints for the Silverpit Crater also show that it is ~12–16 My younger than the pit chain craters[30]. In addition, the morphology of the Silverpit Crater differs substantially from that of the older craters, which do not display a central uplift, concentric faults, and other structural features that are evident in Silverpit. Other crater-forming mechanisms, such as magmatic diapirs[31], erosion by bottom currents[32], or gas escape features[33], also lack the structural characteristics of Silverpit, and thus can be ruled out.

In addition to the seismic characteristics of the crater, the shocked minerals in the adjacent crater floor (within the crater brim) provide strong evidence of an impact origin for Silverpit Crater. Shock metamorphism is uniquely a consequence of hypervelocity impact events and cannot be produced by other terrestrial processes[11]. These shocked minerals are found in a seismic unit equivalent to the crater surface and immediately above it. We note that one of the individual shocked minerals (463 m) observed in the 43/25-1 well is from slightly shallower than the seabed, contemporaneous to the crater floor ( ~12 m shallower than the CF2 reflection). Although we cannot completely rule out that this came from another impact structure close in time and space to Silverpit, we suggest that this was most likely recycled by sedimentary processes in the shallow seabed, sometime after the impact event. Seismic evidence for deltaic clinoforms above the crater (Fig. 2b) demonstrates that this was a dynamic sedimentary environment, with the potential for entrainment and transportation of sediment from the west of the crater.

## Tsunami resurges and secondary craters

The impact would have resulted in a large volume of target material being ejected from the crater floor (Fig. 10). This material would then have been extensively reworked by resurging tsunami waves, based on seismic observations of resurge scars and numerical model observations. The presence of resurge scars at both the base and top (CF1 and CF2 horizons) of the crater-fill sequence at Silverpit implies that the entire transparent package was rapidly deposited in the hours after impact, consistent with observations from other craters[6,34,35]. This crater-fill sequence in Silverpit appears to consist of one seismic package, unlike other, larger marine-impact craters such as Nadir[5,6] or Chicxulub[34]. This might suggest that the crater-fill sequence lacks a thick melt-rich suevite sequence (e.g. ref. [36]) and instead primarily consists of crater resurge.

The resurge likely eroded the crater rims (Fig. 10F), and subsequent seiching in the shallow, partially enclosed basin likely continued for hours or days afterwards (e.g. ref. [37]). High amplitudes associated with this seismic package extends outside of the central crater, at least to the position of the 43/25-1 well where we find evidence for shocked minerals (Figs. 3 and 9). Seiching likely also explains why we find shocked minerals at the site of the well despite this being an oblique impact. Oblique impacts are typically associated with an uprange 'forbidden zone' where ejecta is absent[38,39] or exhibits lower shock pressures. However, in a marine-impact sediment remobilisation over hours and days, likely redistributes at least part of this ejecta more uniformly, including on the uprange side of the crater.

The secondary craters that form on this contemporaneous seabed are consistent with those observed on the moon for a crater of this size, at up to 5% of the size of the parent crater[40]. These indicate that ballistic ejection of large blocks (up to ~5–50-m, depending on impact velocity[40,41]) may have occurred during ejecta emplacement and crater formation. This likely occurred before the resurge stage, as the

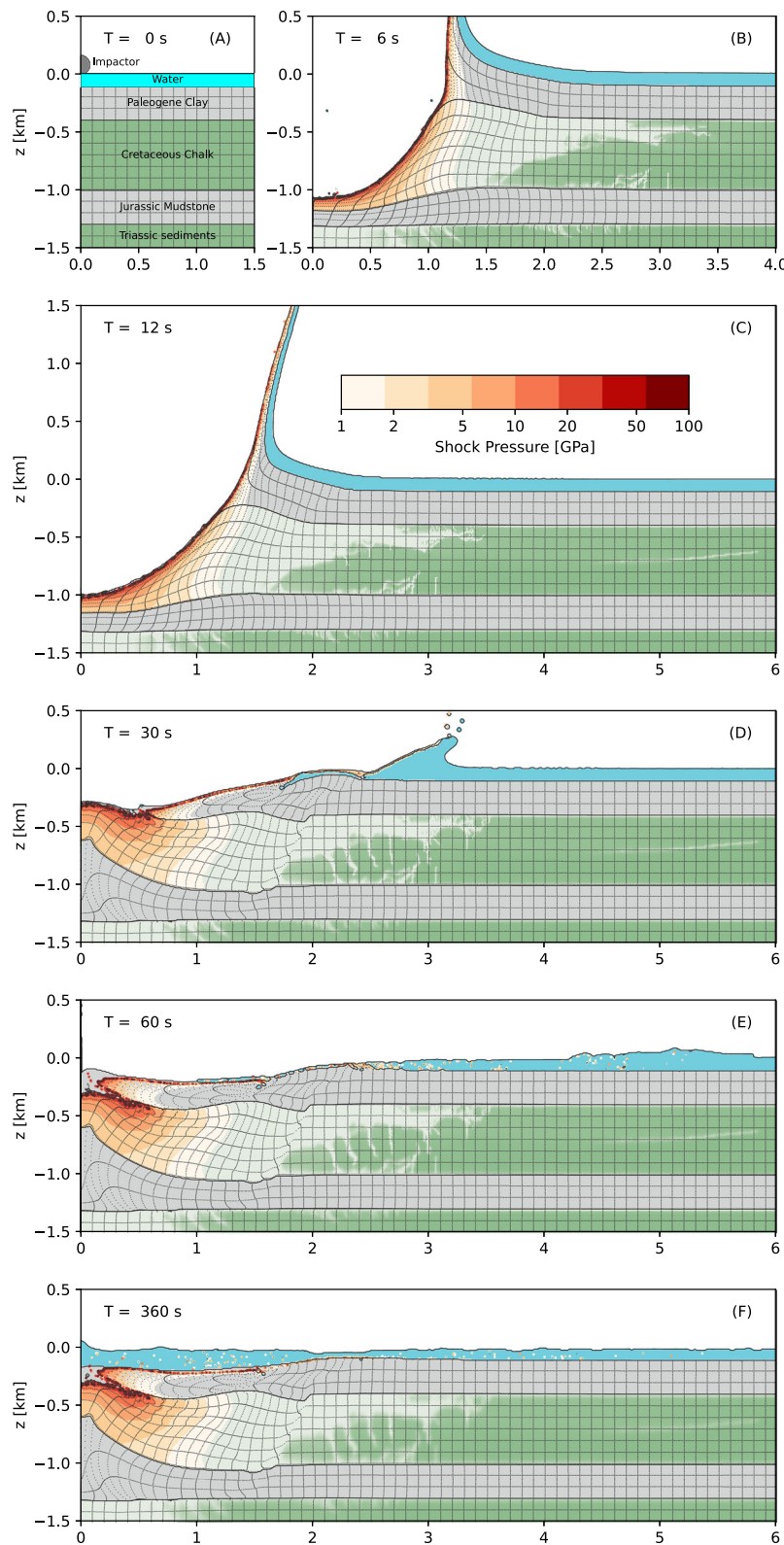

**Fig. 10 | Reconnaissance simulation results for hypervelocity impact of a 160 m diameter rocky asteroid (3300 kg m⁻³ density) at 15 km s⁻¹. A** shows the pre-impact stratigraphic model, followed by the transient crater 6 s (**B**) and 12 s (**C**) after impact. Crater modification and collapse start 30 s after impact (**D**). Crater modification has ceased by 60 s after impact (**E**), but resurge of water into the evacuated cavity continues beyond 120 s (**F**). The model replicates the geometry of the crater, central uplift and moat at the BCU level and Top Chalk, and the envelope of intense brittle damage in the Chalk (light green shading), which shallows with increasing distance from the crater centre. Peak temperatures of 20–30 GPa in the chalk are sufficient to cause devolatilization of the chalk, assuming sufficient shear heating. An animation of the full simulation is included in Supplementary Information (Movie S1).

secondary craters are more pronounced at the CF1 than at the CF2 surface. The secondary craters are also relatively close to the primary crater, possibly because the water depth during the modification stage was shallower (-50 m; Fig. 10E) than before impact, or after the resurge had completed. The resurge, and any subsequent seiche, and associated sediment transport may have partially annealed the secondary craters in the tens of minutes to hours after impact.

Secondary craters are extremely common across other planetary surfaces (e.g. refs. 25,40), but these have rarely been identified on Earth because of the low preservation potential of small terrestrial craters. Secondary craters have recently been proposed in Wyoming[42], possibly in association with a 4.3 km primary crater[43]. The NEP data from Silverpit provide seismic observations of probable secondary impact craters, and direct evidence for secondary craters that can be spatially and temporally correlated to the parent crater.

### Reconstructing impact trajectory

As well as providing robust evidence for an impact origin, the NEP seismic data allow us to constrain the trajectory for the event that generated the Silverpit Crater. There are no pre-existing structures in the target stratigraphy above the Zechstein salt, and the crater likely formed in a shallow (relatively flat) shelf setting[17] so we interpret the structures to be primary impact features.

Curved radial faults in the central uplift provide the best evidence to infer the impact trajectory. Similar faults observed in other sedimentary and marine-target craters, including Upheaval Dome[26], Spider[27], Matt Wilson and Nadir[6], all show concave-uprange fault geometries and downrange verging reverse faults, with an axis of bilateral symmetry used to infer trajectory. These structural observations of inferred asymmetric impacts are validated by numerical models[44,45]. For Silverpit Crater, the deformation patterns at the BCU allow us to plot an axis of bilateral symmetry of approximately 100° from north, indicating an impact from the west-northwest (Fig. 7).

Concentric faults forming in the overlying Paleogene muds and Cretaceous chalk all show normal (extensional) displacement to the west, north and south of the crater (Fig. 5). However, there are conjugate reverse faults to the east. Concentric brim faults are thought to form because of the net-inwards transport of poorly consolidated, water-saturated material in sedimentary and marine impacts (e.g. ref. 24), likely shortly after formation of the central uplift[46]. Reverse faults on one side of the crater in such a setting indicate that the lateral, downrange movement of the seabed resulting from the oblique impact exceeded the reverse flow of material towards the crater during formation of the brim. This also indicates a low-angle impact trajectory from the WNW. Evidence of this lateral movement can also be inferred from the -300 m offset from the centre of the uplift at BCU level relative to the centre of the crater floor, also suggesting greater movement to the ESE in the shallower subsurface (Fig. 2b) during the crater modification stage.

The curved, concave-downrange faults evident at the Top Chalk horizon (Fig. 5b) may provide further, independent evidence of the proposed impact trajectory. These likely form coevally with the concentric faults, accommodating lateral motion of the shallow stratigraphy by strike-slip and reverse slip movement. We suggest that such fault patterns, located exclusively uprange of the crater, could also be used to infer impact trajectory in other craters, in the absence of, or in combination with, the other structural features described above. In this case, an angle normal to the intersecting tangent of both the concentric faults and concave-downrange faults can be used to define a second axis of bilateral symmetry (also -100°; Fig. 5b), parallel to the impact trajectory.

### Carbonate devolatilization in the central uplift

Complex craters typically display a discrete stratigraphic (or central) uplift surrounded by an annular moat, below the crater floor. The central uplift is assumed to be caused by a dynamic weakening process such as acoustic fluidisation[47], as employed in our impact simulations, whereby the rocks or sediments below the crater floor temporarily behave like a non-Newtonian fluid. This allows the substrate below the crater floor to flow vertically upwards and inwards during the early crater modification stage, eventually arresting to form the classic central uplift morphology when the effects of acoustic fluidisation dissipate.

For Silverpit, a classic central uplift is evident at the BCU horizon, but the top chalk displays a more complex and atypical morphology. The central uplift at the top of the Chalk Group is relatively flat, with a pitted texture consisting of large-scale depressions of up to several hundred metres in diameter evident on both the central uplift and the surrounding moat (Fig. 6). The chalk is also significantly thinner below the crater floor than in the surrounding, undeformed sequence. Assuming an original thickness of between 500 m (Fig. 6b) to 600 m (the thickness in 43/25-1; Fig. S2), this corresponds to a volume loss of -0.9–2.2 km$^3$. In addition, there is no central peak observed at the crater floor, as is typically observed in extraterrestrial complex craters.

We suggest that these characteristics are a result of impact-induced thermal decomposition and associated devolatilization that persisted during the crater modification stage. The chalk is made up almost exclusively of calcium carbonate ($CaCO_3$), which undergoes rapid thermal decomposition at temperatures above -750 °C[48] to produce CaO (solid) and $CO_2$ (vapour). Our impact simulations show that the uppermost 100–200 m of the chalk in the 1-km central uplift experiences shock pressures exceeding 25 GPa, similar to the peak pressure required for incipient devolatilization, taking post-shock heating into account[49]. Shock-induced thermal metamorphism is also elevated during oblique impacts, where vaporisation rates and volumes are observed to be higher than vertical impacts[50]. This is not explicitly accounted for in our vertical impact reconnaissance simulations of Silverpit, showing the importance of full 3D, oblique impact simulations for more accurately modelling these effects.

The $CO_2$ produced by full devolatilization corresponds to around 30% of the mass of the original rock volume[51]. This large volume of gas, together with additional water vapour from pore space, would likely have been catastrophically released during the crater modification stage when residual temperatures from shock heating remained high. In this scenario, the mixture of impact melt, water and $CO_2$ could have resulted in an explosive "secondary ejecta plume"[52], with a mixture of gas, molten host rock and rock fragments. This process may explain the chaotic seismic facies in the Paleogene below the crater floor (Fig. 5c) and the extensively pitted top chalk, indicative of fluid escape features (e.g. ref. 53), and possibly the amorphous carbonate samples in some cuttings at 494 m in well 43/25-1. Pits of similar dimensions have been observed in impact craters on Mars, and are also inferred to be caused by devolatilization after impact[54]. Our data strengthen that hypothesis by revealing the subsurface morphology of these structures.

These surface pits below the crater floor are morphologically quite different from the secondary craters that formed beyond the crater rim, the latter of which are much smaller and more widely distributed. Nevertheless, an alternative genetic process for the secondary craters could be degassing, perhaps with fluids migrating along the complex fault structures from an area below the crater floor that experienced extreme shock pressures and temperatures. However, the clear morphological differences and the fact that the surface pits in the central nested crater show strong evidence of 'bottom-up' disturbance that is absent in the secondary craters lead us to conclude that the genetic processes are quite different for these features. Both sets of features are observed geophysically in Silverpit Crater and require further modelling and perhaps recovery of core samples by scientific drilling to fully test these hypotheses.

## Methods

The Silverpit Crater has been assessed on the previously interpreted PGS Southern North Sea MegaSurvey and a high-resolution 3D seismic dataset acquired in 2022 on behalf of the Northern Endurance Partnership (NEP). This Greater NEP 3D seismic survey was acquired in water depths of 21–93 m. Acquisition parameters include 4 × 400 cubic inch sources and 9 × 3 km streamers with a separation distance of 50 m (ref. 31). The shot interval was 6.25 m, with a sample rate of 2 ms. Bin dimensions are 6.25 x 6.25 m for acquisition and 12.5 x 12.5 m for processing, with a nominal fold of 80. The record length for the survey is 4000 ms. Data were processed using a proprietary Pre-Stack Depth Migration (PSDM) sequence including full-waveform inversion (FWI).

The Pre-stack Depth Migrated (PSDM) data, acquired and processed using modern techniques, provides a significant uplift in data quality compared to previously available post-stack time-migrated datasets, which focused on pre-Zechstein imaging. Notably, the NEP data has greater frequency bandwidth, improved fault imaging and fewer multiples across the zones of interest (Fig. 2 and S1). The combined datasets provide 100% coverage of the crater (within the rim, sensu[20]) and 95% coverage of the entire damage zone associated with the crater. The NEP data were interpreted in the time rather than depth domain, for seamless interpretation with older Megamerge data, in particular when correlating the surfaces with the 43/25-1 well, which is beyond the limit of the NEP data. Data in the time domain were generated using the same velocity model as that used to depth-migrate the PSDM data.

Seismic interpretation was carried out using Schlumberger Petrel 2020 software, including horizon mapping, structural element mapping, velocity analysis and attribute analysis.

The 43/25-1 well was drilled in 1985 by British Gas. The petrophysical data and drilling reports were made available by the North Sea Transition Authority (NSTA) (https://ndr.nstauthority.co.uk/). The well was drilled for a deeper (Palaeozoic) target with a relatively limited log suite in the shallow section. The samples in the interval of interest are from drill cuttings, with some inherent possibility of contamination from borehole collapse during drilling, which could lead to shallower (younger) samples mixing with those from the assigned depths. Caliper log data and biostratigraphic results suggest that such contamination is absent or minimal. Nevertheless, there is some outstanding depth uncertainty due to mixing in the drilling mud in the borehole, between the subsurface and the shakers on the drilling rig, where the samples are collected.

The high-amplitude reflections (CF1 and CF2) interpreted to represent the seabed contemporaneous to the crater floor were correlated northwest of the crater rim, intersecting the 43/25-1 well at 502 and 522 ms TWT. Time/depth pairs derived from borehole seismic data (checkshots) were available for 43/25-1, but the shallowest measurement available was at 914 m, within the Chalk Group. A time-depth model for the well based on this checkshot data alone suggests that the CF1 reflection intersects the well at ~580 m MD (measured depth) (Fig. S2). However, extrapolation of this trend to the surface results in a significant overestimation of shallow velocities. To constrain this further, four shallow checkshots from within the Cenozoic stratigraphic section were spliced in from the 43/24-3 well, 5 km to the northeast. A time-depth model incorporating these checkshots with those already measured at 43/25-1 results in a shallower estimate of the CF1 horizon, at ~540 m. The velocity model was further refined by generating a synthetic seismogram at the well location, using wireline log datasets. The manual correlation of this synthetic to adjacent seismic traces from the PGS MegaSurvey resulted in a shallower estimate of the CF1 crater floor horizon, at ~495 m. We consider this last value to be the most likely depth for the crater-floor equivalent seabed.

The well-tie shows a good match at all key stratigraphic levels, and a fair character match across the Paleogene section (Fig. S3). There is some uncertainty on the seismic well-tie and exact intersection with the crater due to lack of checkshot data at the crater level and limited sonic data above the crater to incorporate low frequency components into the synthetic.

For calcareous nannofossil biostratigraphic analysis, 31 subsamples of the most representative lithologies were carefully selected from cuttings samples across the 183–610 m interval. Samples were prepared using the simple smear slide technique for nannofossil observation[55]. Examination of smear slides was completed using a transmitted-light microscope (Zeiss AxioScope at x1250 magnification) under cross-polarised light (XPL). Age determinations are based on the calibrated bioevents for low and middle latitudes[56]. Nannofossil taxonomy follows[57–60] as compiled in the online Nannotax 3 database.

For the petrographic analysis, 27 cutting samples were selected from the 421–603 m interval, covering the full range of crater floor equivalent estimates for the well. Cuttings were mounted on glass and embedded in epoxy, then the aggregates were sawed and polished to 30 μm thickness. Quartz grains in these samples were visually inspected for evidence of shock lamellae or other shock metamorphic features. Candidate shocked minerals were analysed using a scanning electron microscope with energy dispersive X-ray spectroscopy (SEM/EDX) to confirm mineral species and for crystallographic indexing using a U-stage. Estimated pressures for shock features are from ref. 10.

Impact simulations were performed using the iSALE shock physics code[61], which solves the equations for conservation of mass, momentum, and energy using a finite-difference approach based on the SALE hydrocode. The specific physical response of a material to impact is approximated using an equation of state to describe volumetric response and a shear strength model. Impactor and target materials were approximated using the closest available analogue material model.

A simplified pre-impact target structure including the major sedimentary divisions was constructed based on seismic and drill core results from the area. A spatial resolution of a 10-m per grid cell was used to resolve the impactor and target to a 6-km radius and 1.5 km depth, with a zone of coarser resolution extending to over 13 km depth and 17 km radius. Beneath the seafloor was 300 m of ductile clay overlying 600 m of brittle chalk, and a 300-m layer of mudstone. A further layer of chalk extended beneath this to the bottom of the mesh. The sedimentary sequence was overlain by 100 m of water. ANEOS-derived equation of state tables for calcite[62] and quartz[63] were used to represent the chalk and clay/mudstone layers, respectively. Water-filled porosity in the sediments was neglected. To produce the enhanced mobility of rock masses during impact crater collapse (c.f. [47]), we incorporated the Acoustic Fluidisation Block Model parameters used for carbonates[64]. As the choice of rock strength and material weakening parameters has a substantial effect on the final crater form and subsurface formation, we explored a range of potentially suitable values. The parameters used in the reconnaissance simulation that best replicated observations are given in Supplementary Information Table S2. We note the large difference in limiting strength ($Y_m$) between the clay/mudstone and the chalk, which gives the clay/mudstone its weaker, more ductile behaviour, as well as the relatively low damaged material friction coefficient ($\mu_d$) to account for water saturation of the sediments.

To produce a crater the size of the internal Silver Pit structure (~3.2 km diameter), the impactor was approximated as a 160-m diameter sphere striking the surface at a speed of 15 km s$^{-1}$. A dunite equation of state and strength model was used for the projectile[65], with a reference density of 3300 kg m$^{-3}$. A vertical impact was assumed for

computational expediency in these reconnaissance simulations. Future work will explore the effect of the impact angle, given the evidence for oblique impact in the structural deformation. These impactor properties are within the range expected for terrestrial meteors[66].

## Data availability

All data needed to evaluate the conclusions in the paper are present in the paper and/or the Supplementary Information. Seismic data are from NEP under a confidentiality agreement and remain the property of that organisation. Enquiries about data access should be directed to NEP (info@nephccp.co.uk).

## Code availability

Scientists interested in using or developing iSALE should see https://isale-code.github.io/index.html for a description of application requirements.

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

## Acknowledgements

We thank the Northern Endurance Partnership and NSTA for providing access to 3D seismic data for this project, and the National Data Repository (NDR) operated by the North Sea Transition Authority. We would like to acknowledge the British Geological Survey's National Geomaterials Repository (NGR) technicians and staff who enabled access to samples SSK151388-151411 & SSK151420-151427 from the 43/25-1 well, which has supported this research [NGR Reference: IDA290499]. We also thank Schlumberger for the provision of an academic license of Petrel 2020 for seismic interpretation. Thanks also to Amy Gough and Catherine Ross for reviewing a draft of this manuscript prior to submission. UN, TDJ and SG acknowledge NERC grant number NE/W009927/1. This is University of Texas Institute for Geophysics Contribution #3978 and Centre for Planetary Systems Habitability Contribution #0082. GSC acknowledges support from STFC grant ST/S000615/1.

## Author contributions

U.N. was responsible for conceptualisation, wrote the original draft and carried out subsequent editing. I.d.J.A., A.G. and J.F. carried out seismic data interpretation and contributed to writing and editing the manuscript. T.D.J. carried out analytical work (nannofossils), G.C. and V.B. carried out numerical modelling, T.K. carried out analytical work (petrography and SEM analysis) and helped with structural interpretation. R.P. was involved in conceptualisation during the early stage of the project. S.G. contributed to data interpretation. All authors contributed to writing and editing the manuscript.

## Competing interests

The authors declare no competing interests.
