## [Transparent Peer Review file · Nature Communications]

The silver bullet - evidence for a hypervelocity impact origin for the Silverpit Crater

Corresponding Author: Dr Uisdean Alasdair Nicholson

Version 0:

Reviewer comments:

Reviewer #1

(Remarks to the Author)

The authors have conducted a thorough study of new data from the Silverpit Crater, the origin of which has been debated. The paper presents new evidence that supports an impact origin for the crater. I recommend that paper be published after addressing the following remarks.

Introduction Line 29 – space between “than” and “a”.

In the introduction and results, the authors have stated that the diameter of this suspected impact structure is reported to be 3 km. I would recommend adding an explanation somewhere in the text about how this diameter is defined (either by previous work – and then state how that was redefined from the original 20km, or in the current work and what criteria were used to set the diameter at 3km – are these arguments new/different, or in agreement with the previously defined 3km boundary?) A crater fill package is mentioned a few times throughout the text, and there is some interpretation about its deposition. Is there any indication as to the material of this crater fill? It might be useful to elaborate a little bit about what is generally expected in a crater fill deposit in a complex crater, and how the observations/simulations in this study might support the interpretation of this unit as a crater fill deposit.

Reconstructing impact trajectory –

Lines 1- 5: the authors have mentioned several other craters with sedimentary targets where fault systems exist. It is unclear whether impact trajectories at these sites were inferred solely from fault systems, or if there were other lines of independent evidence. It is also unclear whether these other craters are analogous to the Silverpit crater (other than being sedimentary targets – are these also marine impacts?)

Is there any indication of pre-existing fault systems, deformation or zones of weaknesses in the target lithology in the Silverpit crater area that could affect the manifestation of impact-related faulting? It is important to consider potential pre-existing features while reconstructing the impact trajectory from structural evidence.

Results – Seismic observations

Line 30 – the pits are interpreted as secondary craters, but the authors have not provided convincing evidence for that. Secondary craters have not been definitively identified on Earth and so it is understandable that evidence of such features is difficult to identify from seismic data only. However, it would be helpful to discuss more about why the authors have interpreted these features as secondary craters other than just their size.

Additionally, there are pitted textures identified in a different part of the crater. Is there any relationship between the “secondary craters” and these pits on the inferred central uplift? From the evidence presented in the paper, it seems that the only main difference between the pits and the “secondary craters” may be their size. How do you distinguish between these two features – could they be formed by the same mechanism?

I suggest that the authors add a schematic figure that visualizes the crater morphology and the evidence of impact observed in this study – it will be useful to compile all the observations into a single figure.

Figure 2 –

“A” and “B” labels are missings.

Figure 2A shows a profile across NNW-SSW, but the caption says the profile is across NNW-SSE.

Reviewer #2

(Remarks to the Author)

The manuscript from Nicholson et al. titled "The silver bullet - new evidence for a hypervelocity impact origin for the Silverpit Crater" is a well-rounded work trying to confirm the impact origin of the controversial Silverpit crater in the North Sea. This work found new evidence of shock metamorphism in quartz and K-feldspar from drill cores, and presented new seismic data of the crater, and ran new numerical simulations to model the impact events. Overall, the impact origin of Silverpit Crater is supported by the evidence shown from this work. The publication of this work will help settle the debate in impact cratering community and provide new prospective for future discoveries of buried impact craters.

The methods are described in detail, the data analysis and interpretation of the drill cores are decent and meet the standards in the field. The modeling code used parameters based on the seismic data and drill core observations. The figures are well-labeled and clear to read. The writing is smooth, and the logic flow is easy to follow.

A few minor comments:

1. How many drill wells are there? Why so few? Any more planned? Is there any other shock metamorphism evidence observed in the well and checkshots? Any impact melt? Diaplectic glass? Any leftover from the impactor?
2. Figure 6c: what is that obvious resolution change in the left to middle lower part of the image? Like it looks suddenly a bit blurry?
3. For easier to read from shallow to deep depth, I would suggest re-arranging the pictures in Figure 9: A, B, C in one row labeled 463.3m, and D, E, F in the second row labelled 493.8m; same suggestion for Figure S4, the 494m on top and 500m in the bottom. Also, the label A-F in Figure 9 is capital while in caption is a-f.
4. Is it possible to identify and label other minerals in SEM image Figure 9A? Any element mapping? And if the quartz and K-feldspar can be seen from the SEM image, please consider outlining them. Also, in Figure 9E, label "Kfsp" should be placed on the crystal, or use an arrow, otherwise it looks like the black area is Kfsp.
5. Overall, if the silver bullet is the petrographic evidence of shock metamorphism, I would suggest having a bigger proportion of it in the manuscript (better balance between seismic data, petrographic and modeling), e.g. make figure 9 into a big and beautiful one and move some of the description from supplementary into the main text.

Good job and good luck!

Reviewer #3

(Remarks to the Author)

The manuscript "The silver bullet - new evidence for a hypervelocity impact origin for the Silverpit Crater" by Nicholson et al. reexamined the Silverpit Crater in the North Sea. Analyzing the high-resolution 3D seismic data, they confirmed that Silverpit is actually a crater, which was disputed previously. They also found that drilled samples from the 43/25-1 well present a dearth of organics at the same depth level of the crater floor, supporting an energetic event, an impact had happened. Their impact modelling, which successfully reproduces the shape of observed craters, also supports their confirmation. As such, this paper investigated the Silverpit Crater from multiple points of view. I am sure that many researchers working in the various fields will be interested in this paper. Nevertheless, I feel like some additional discussions would be valuable and enhance future research on the formation of this crater especially from the viewpoint of oblique impacts.

1. Shocked grains from drilled sample

The authors identified two grains metamorphosed via the highly shocked pressure. They argued that the grains are close enough to the crater thus supporting the impact origin. The 43/25-1 well is located 1 km north-west of the crater rim (Page 5, Line 35), and they mentioned "Shocked material deposited in the clay" (Page 7, Line 20). Therefore, I suppose those grains could be an impact ejecta. If correct, please clarify the experienced pressure of impact ejecta from their impact model. Although the impact result shows the experienced pressure of each layer (Fig. 10), it is hard to identify the ejecta's pressure as written in the abstract ("indicating shock pressures of ~10-13 GPa, consistent with results from our numerical models"). Also, how can they exclude the possibility that the shocked grains are delivered by other impacts in a similar era? If I misunderstood something, please correct me.

2. Effect of oblique impact

They discussed the impact trajectory using the deformation around the crater. While they further argued that the impact angle would be low, the discussions about the effect of (low-angle) oblique impacts are less. I acknowledge their great work on the reconnaissance impact simulations and the oblique impact simulations are beyond their focus. Nevertheless, it would be worth discussing its general effect. For example, the amount of impact ejecta in the uprange direction is less than those in the downrange direction. Moreover, there is a V-shaped zone of no impact ejecta in the uprange direction (e.g., Gault and Wedekind 1978, Luo et al. 2022). Since the drilled site is in the uprange direction of the suggested impact trajectory, the shocked ejecta taken from the 43/25-1 well might be the rare case, if my above comment is correct. Alternatively, even if I am wrong, the shock pressure in the uprange direction tends to be lower than that in the downrange direction (e.g., Collins et al. 2022). As such, more discussions could be better to link between the shocked minerals and impact models in terms of oblique impacts.

3. Pits; secondary craters and/or devolatilization

The authors found that several pits of ~150 m located at the crater brim, and interpreted those pits as secondary craters (Page 4, Line 26-32 and Fig. 3). While they suggested that the size ratio between primary and secondary craters is comparable to other planets, I am curious if this still holds for the Silverpit Crater case. Because the ocean affected the crater formation. They discussed that the secondary craters could be free from the resurge tsunami (Page 8, Line 28-35), which might be true. However, the impact movie does not indicate this clearly, rather the formation of secondary craters might be also influenced by the ocean. On the other hand, they interpreted other large pits of 50-500 m diameter located around the

crater center as a different origin, devolatilization (Page 5, 13-21 and Fig. 6). Could the secondary craters also have been formed via devolatilization? Or can they exclude this and other possibilities and validate these are sure to be the secondary craters? Please also clarify the difference/similarity between these pits and their origins.

Additionally, I think an ejecta of 50-100 m seems a too large block (Page 8, Line 29), given that the impactor size is 160 m in diameter and the secondary crater is up to 150 m. How do the authors estimate this block size?

4. Formation of the concentric faults

As Stewart and Allen (2002) mentioned that the Silverpit Crater is a multi-ringed impact structure, the concentric faults on the chalk (Fig. 5) remind me of multiring basins on icy satellites (Valhalla on Callisto, Tyre and Callanish on Europa). I agree that "the rheological contrasts between different rock types" (Page 5, Line 3) could be the key. At the same time, I am wondering if this contrast could fit with the suggested mechanism of multiring basin formation (Melosh and McKinnon 1978, McKinnon and Melosh 1980); i.e., can the chalk over the mudstone correspond to the lithosphere over the asthenosphere? Are there any features shown in the impact model? Please comment on this.

Here are some other minor comments.

Page 6, Line 11: Fig. S5 is missing, or just a typo of S4?

Page 9, Line 43: How about comparing with terrestrial complex craters, not extraterrestrial ones?

Page 12, Lines 4-5: Please state the mesh size. It should be fine enough to resolve the layer and the diameter of 160 m (Line 20). Relating to this, adding the impactor's properties in Table S2 might be better.

References:

Collins et al. 2022: <https://doi.org/10.1038/s41467-020-15269-x>

Gault and Wedekind 1978: <https://ui.adsabs.harvard.edu/abs/1978LPSC....9.3843G/abstract>

Luo et al. 2022: <https://doi.org/10.1029/2022JE007333>

McKinnon and Melosh 1980: [https://doi.org/10.1016/0019-1035\(80\)90037-8](https://doi.org/10.1016/0019-1035(80)90037-8)

Melosh and McKinnon 1978: <https://doi.org/10.1029/GL005i011p00985>

Reviewer #4

(Remarks to the Author)

This a very well written paper which clearly demonstrates that the Silverpit structure is an impact crater. The seismic data are of very high quality and provide convincing evidence for the impact origin even without the complementary shocked mineral evidence. The authors might want to consider the following comments prior to a decision on acceptance.

GENERAL COMMENTS

A crater diameter of 3 km is close to the limit separating simple craters from complex craters. The presence of a central uplift in the seismic data suggests the Silverpit crater falls into the category for complex craters and that a transient crater would have been present directly after impact that would have been somewhat smaller than the present day final diameter. Some discussion on the scaling relationships between simple and complex craters and the implications of crater type on the seismic interpretation would have been useful.

According to the authors PSDM was applied to the seismic data. However, all vertical sections through the seismic volume are in TWT. Is this a mistake or have the seismic data been converted back from depth to time?

Reflector vs. reflection: These terms seem to be used interchangeably in the MS. In my view, a reflection is something which is observed on a seismic section while a reflector is something which is present in the sub-surface. You may want to consider using these definitions in the Results and Discussion sections.

SPECIFIC COMMENTS

Pg 3, lines 13-15: "Although the structure was still only partially imaged, these data allowed the crater rim to be redefined as a much smaller 3 km in diameter, with some evidence of radial faults on one side of the central uplift." This sentence is confusing, is the crater 3 km in diameter or is it 3 km smaller in diameter than the 20 km mentioned earlier on line 7? From Figure 1 and further reading of the MS, I see it is the former, but it would be better to be more clear here.

Figure 2b: What does the "300 m offset" label in the figure refer to? It is explained later in the discussion, but please explain in figure caption.

Pg 3, line 29: Change "rather than a decision" to "rather than a decision".

Figure 5: It would be useful to be consistent with the colors for the faults. For example; green – reverse, red – normal; for this

figure and the others in the MS. Also what does the dotted line in c) represent?

Pg 6, lines 29-30: "in different domains of the crystal, which might that the grain additionally shows Dauphine-type twinning" Something is missing in this sentence. Also, you could already here note that the presence of shocked quartz above the CF2 reflector will be discussed later.

Figure 8: CF1 and CF2 appear to be incorrectly labeled in the AI panel.

Pg 7, lines 1-3: It appears you are assuming the impact to be vertical, not at an angle. I also interpret this to be the case looking at Figure 10 (symmetrical). You should state this here. I note later that you write the modeling assumed a vertical impact, but better to make the point at this stage.

Pg 8, line 24: Change "10.f" to "10f".

Christopher Juhlin, Uppsala Univeristy
2025-04-09

Version 1:

Reviewer comments:

Reviewer #1

(Remarks to the Author)

The authors present convincing new evidence and a detailed study of the Silverpit structure, a crater with a disputed impact origin. The new results are significantly in favor of an impact origin for the structure, and will confirm a new addition to the terrestrial impact cratering record. The methods, results and discussion are sound, and well presented, with high quality figures. The authors have satisfactorily addressed the suggested revisions from the previous version, and I recommend the manuscript be accepted for publication.

Best Regards,
Dr. Neeraja Chinchalkar

Reviewer #2

(Remarks to the Author)

I'm happy with the revised manuscript and I would love to see it published.

Reviewer #3

(Remarks to the Author)

I thank the authors for fully addressing all of my comments in the revised manuscript. I recommend this paper for publication.

Reviewer #4

(Remarks to the Author)

I happy with the revised manuscript. I only have the following two comments:

1. Line 480: I think "in particular to correlated the surfaces with the 43/25-1 well" should be "in particular when correlating the surfaces with the 43/25-1 well"

2. Line 481: I think the authors should add a sentence stating what velocity field was used to convert the PSDM image to a PSTM image. I assume it was the velocity field used for PSDM.

Point-by-point response

We thank the four reviewers for their excellent comments and questions. These have all been addressed and we provide point-by-point responses to each comment below. Reviewers' comments are in black and our responses are in blue.

REVIEWER COMMENTS

Reviewer #1 (Remarks to the Author):

The authors have conducted a thorough study of new data from the Silverpit Crater, the origin of which has been debated. The paper presents new evidence that supports an impact origin for the crater. I recommend that paper be published after addressing the following remarks. *We thank the reviewer for these positive comments.*

Introduction Line 29 – space between “than” and “a”. *Done*

In the introduction and results, the authors have stated that the diameter of this suspected impact structure is reported to be 3 km. I would recommend adding an explanation somewhere in the text about how this diameter is defined (either by previous work – and then state how that was redefined from the original 20km, or in the current work and what criteria were used to set the diameter at 3km – are these arguments new/different, or in agreement with the previously defined 3km boundary?) *The section in the introduction has been expanded to clarify that the crater was still interpreted by Stewart *et al.* (2005) to be at the Top Cretaceous, and at 8 km in diameter (the 3 km in diameter referred to the diameter of the crater immediately after impact, as they also propose post-impact collapse to the present geometry). We have also expanded the description of the rim in the Results section to explain this more clearly, and how it differs from that described in this paper in particular.*

A crater fill package is mentioned a few times throughout the text, and there is some interpretation about its deposition. Is there any indication as to the material of this crater fill? It might be useful to elaborate a little bit about what is generally expected in a crater fill deposit in a complex crater, and how the observations/simulations in this study might support the interpretation of this unit as a crater fill deposit. *We think that this crater fill is likely dominated by resurge-deposits – sorted breccia or clastic sediment transported back into the crater during the resurge event. We have added a couple of additional sentences to the relevant section in the discussion (Tsunami resurge and secondary craters), including the fact that we do not expect to see significant melt lenses/suevite, based on the seismic observations.*

Reconstructing impact trajectory –

Lines 1- 5: the authors have mentioned several other craters with sedimentary targets where fault systems exist. It is unclear whether impact trajectories at these sites were inferred solely from fault systems, or if there were other lines of independent evidence. It is also unclear whether these other craters are analogous to the Silverpit crater (other than being sedimentary targets – are these also marine impacts?). *These observations are inferred to represent oblique impacts based on numerical model results which replicate this structure. This has been*

clarified in the text. These craters are not all marine craters, but that is not critical here – such structures can also occur in crystalline rock as well.

Is there any indication of pre-existing fault systems, deformation or zones of weaknesses in the target lithology in the Silverpit crater area that could affect the manifestation of impact-related faulting? It is important to consider potential pre-existing features while reconstructing the impact trajectory from structural evidence. There is no indication of pre-existing structures in the target stragrophy, at least not above the Zechstein salt, which is well below the maximum depth of deformation. We have added a sentence to clarify this at the beginning of this section.

Results – Seismic observations

Line 30 – the pits are interpreted as secondary craters, but the authors have not provided convincing evidence for that. Secondary craters have not been definitively identified on Earth and so it is understandable that evidence of such features is difficult to identify from seismic data only. However, it would be helpful to discuss more about why the authors have interpreted these features as secondary craters other than just their size. We acknowledge that it is challenging to fully document the evidence for this, in a short paper format. However, we think that the evidence for these as secondary craters is compelling, including: (1) no seismic evidence for ‘bottom-up’ deformation below the craters, unlike the pits in the central uplift (see below); (2) these are consistent with the expected size for secondary craters in extraterrestrial structures; (3) a lack of a compelling alternative origin for these (see below) and (4) consistent size (parent to secondary crater ratio) as observed on secondary craters on the moon. We have added some additional explanation of this in the text, but we have also acknowledged alternative origins for these (devolatilization of water from clastic sediments). See also comments in response to Reviewer 3, below.

Additionally, there are pitted textures identified in a different part of the crater. Is there any relationship between the “secondary craters” and these pits on the inferred central uplift? From the evidence presented in the paper, it seems that the only main difference between the pits and the “secondary craters” may be their size. How do you distinguish between these two features – could they be formed by the same mechanism? The two sets of features are morphologically quite different, and occur in areas with very different pressure/temperature conditions during impact, so we think that these are genetically different. The features interpreted as secondary craters, outside the crater, are widely dispersed, extending at least 3 km radially from the centre of the crater. The subsurface temperature in this area is, according to our numerical models, not sufficient to generate major devolatilization of carbonates in this area. By contrast, those around the central uplift and crater floor are associated with significant seismic disturbance of the chalk and Paleogene clastics (so evidence of ‘bottom-up’ deformation), and are much larger and more tightly clustered. We have clarified this in the manuscript, including a new paragraph at the end of the Discussion acknowledging that this needs to be tested further. We have also made a new version of Fig. 6a to display the central pits (that we infer are caused by devolatilization) more clearly.

I suggest that the authors add a schematic figure that visualizes the crater morphology and the evidence of impact observed in this study – it will be useful to compile all the observations into a single figure. This would be good to add but we have already reached the figure limit (10 max)

for *Nat Comms*. This can be added in future papers on this structure though.

Figure 2 –

“A” and “B” labels are missings. These labels have been added.

Figure 2A shows a profile across NNW-SSW, but the caption says the profile is across NNW-SSE. The caption has been changed to NNW-SSW.

Reviewer #2 (Remarks to the Author):

The manuscript from Nicholson et al. titled “The silver bullet - new evidence for a hypervelocity impact origin for the Silverpit Crater” is a well-rounded work trying to confirm the impact origin of the controversial Silverpit crater in the North Sea. This work found new evidence of shock metamorphism in quartz and K-feldspar from drill cores, and presented new seismic data of the crater, and ran new numerical simulations to model the impact events. Overall, the impact origin of Silverpit Crater is supported by the evidence shown from this work. The publication of this work will help settle the debate in impact cratering community and provide new prospective for future discoveries of buried impact craters. We thank the reviewer for these positive comments and agree that this should resolve the debate around the origin of Silverpit.

The methods are described in detail, the data analysis and interpretation of the drill cores are decent and meet the standards in the field. The modeling code used parameters based on the seismic data and drill core observations. The figures are well-labeled and clear to read. The writing is smooth, and the logic flow is easy to follow.

A few minor comments:

1. How many drill wells are there? Why so few? Any more planned? Is there any other shock metamorphism evidence observed in the well and checkshots? Any impact melt? Diaplectic glass? Any leftover from the impactor? There are only two exploration wells drilled within the vicinity of the crater, within the outer limit of the damage zone, both shown on figure 1. These are old oil exploration wells, drilled long before an impact origin of the structure was proposed. The 43/24-3 well sits within the shallow seismic sequence that is almost completely obscured by multiples, thus defining the depth of the crater floor is very difficult (and it may have been entirely eroded by the Quaternary glaciation). Even the 43/25-1 well sits outside of the crater rim, so no impact melt, etc., although this could potentially be present in small amounts in the ejecta. And there are only cuttings (rock chips) available from every 15 feet or so, so the chances of finding impact evidence was very small, *a priori*. We note that we are now considering proposing a scientific ocean drilling expedition to drill and recover continuous core from this crater in a more optimal location in the coming years, to test some of the hypotheses that we have developed in this paper. This may allow some of these impactites to be recovered as well.

2. Figure 6c: what is that obvious resolution change in the left to middle lower part of the image? Like it looks suddenly a bit blurry? This is the boundary between the NEP3D and PGS Megamerge survey. This was mentioned in the figure caption, but is now added to the figure for greater clarity.

3. For easier to read from shallow to deep depth, I would suggest re-arranging the pictures in Figure 9: A, B, C in one row labeled 463.3m, and D, E, F in the second row labelled 493.8m; same suggestion for Figure S4, the 494m on top and 500m in the bottom. Also, the label A-F in Figure 9 is capital while in caption is a-f. This has been done.

4. Is it possible to identify and label other minerals in SEM image Figure 9A? Any element mapping? And if the quartz and K-feldspar can be seen from the SEM image, please consider outlining them. Also, in Figure 9E, label "Kfsp" should be placed on the crystal, or use an arrow, otherwise it looks like the black area is Kfsp. Figure 9 has been rearranged and labels added to highlight the shocked minerals.

5. Overall, if the silver bullet is the petrographic evidence of shock metamorphism, I would suggest having a bigger proportion of it in the manuscript (better balance between seismic data, petrographic and modeling), e.g. make figure 9 into a big and beautiful one and move some of the description from supplementary into the main text. We added some text to this section, but have had to be conservative with this, as we still have to keep the overall paper relatively short for *Nature Communications*. Likewise, we need to keep figure sizes as small as possible, without compromising legibility or clear presentation of the necessary evidence.

Good job and good luck! Thank you!

Reviewer #3 (Remarks to the Author):

The manuscript "The silver bullet - new evidence for a hypervelocity impact origin for the Silverpit Crater" by Nicholson et al. reexamined the Silverpit Crater in the North Sea. Analyzing the high-resolution 3D seismic data, they confirmed that Silverpit is actually a crater, which was disputed previously. They also found that drilled samples from the 43/25-1 well present a dearth of organics at the same depth level of the crater floor, supporting an energetic event, an impact had happened. Their impact modelling, which successfully reproduces the shape of observed craters, also supports their confirmation. As such, this paper investigated the Silverpit Crater from multiple points of view. I am sure that many researchers working in the various fields will be interested in this paper. Nevertheless, I feel like some additional discussions would be valuable and enhance future research on the formation of this crater especially from the viewpoint of oblique impacts. We thank reviewer 3 for these positive comments, and address their concerns below.

1. Shocked grains from drilled sample

The authors identified two grains metamorphosed via the highly shocked pressure. They argued that the grains are close enough to the crater thus supporting the impact origin. The 43/25-1 well is located 1 km north-west of the crater rim (Page 5, Line 35), and they mentioned "Shocked material deposited in the clay" (Page 7, Line 20). Therefore, I suppose those grains could be an impact ejecta. If correct, please clarify the experienced pressure of impact ejecta from their impact model. Although the impact result shows the experienced pressure of each layer (Fig. 10), it is hard to identify the ejecta's pressure as written in the abstract ("indicating shock pressures of ~10-13 GPa, consistent with results from our numerical models"). The shock pressure in the ejecta was stated previously in the Results section on impact modelling, but we have re-organised this for greater clarity, and to explicitly state that shock pressures in the ejecta reach up to 30 GPa.

Also, how can they exclude the possibility that the shocked grains are delivered by other impacts in a similar era? If I misunderstood something, please correct me. This can never fully be excluded, but shocked grains are so rare in nature that other impacts are not necessary to invoke here. We know that these grains are at, and stratigraphically slightly above, the Eocene

seabed just outside the crater. This evidence is much more robust than for other confirmed terrestrial craters. For example, the Hiawatha impact structure appears as a confirmed crater in impact databases (e.g. Impact Earth: <https://impact.uwo.ca/map/>), confirmed by shocked minerals in glacial outwash deposits several kms away from the crater, with no stratigraphic correlation. Nevertheless, we added a sentence to this discussion to acknowledge that derivation from another impact event is a possibility (albeit unlikely).

Kenny, G.G., Hyde, W.R., Storey, M., Garde, A.A., Whitehouse, M.J., Beck, P., Johansson, L., Søndergaard, A.S., Bjørk, A.A., MacGregor, J.A. and Khan, S.A., 2022. A Late Paleocene age for Greenland's Hiawatha impact structure. *Science Advances*, 8(10), p.eabm2434.

2. Effect of oblique impact

They discussed the impact trajectory using the deformation around the crater. While they further argued that the impact angle would be low, the discussions about the effect of (low-angle) oblique impacts are less. I acknowledge their great work on the reconnaissance impact simulations and the oblique impact simulations are beyond their focus. Nevertheless, it would be worth discussing its general effect. For example, the amount of impact ejecta in the uprange direction is less than those in the downrange direction. Moreover, there is a V-shaped zone of no impact ejecta in the uprange direction (e.g., Gault and Wedekind 1978, Luo et al. 2022). Since the drilled site is in the uprange direction of the suggested impact trajectory, the shocked ejecta taken from the 43/25-1 well might be the rare case, if my above comment is correct.

Alternatively, even if I am wrong, the shock pressure in the uprange direction tends to be lower than that in the downrange direction (e.g., Collins et al. 2022). As such, more discussions could be better to link between the shocked minerals and impact models in terms of oblique impacts. This is a point we had considered but not developed in detail in the manuscript. Ejecta is indeed typically distributed downrange and lateral to the crater following oblique impacts, with an uprange “forbidden zone” of no ejecta. However, in marine impacts, the tsunami resurge and subsequent seiching redistributes this sediment, including in the uprange direction. We have extended this section by adding several sentences to explain this process.

3. Pits; secondary craters and/or devolatilization

The authors found that several pits of ~150 m located at the crater brim, and interpreted those pits as secondary craters (Page 4, Line 26-32 and Fig. 3). While they suggested that the size ratio between primary and secondary craters is comparable to other planets, I am curious if this still holds for the Silverpit Crater case. Because the ocean affected the crater formation. They discussed that the secondary craters could be free from the resurge tsunami (Page 8, Line 28-35), which might be true. However, the impact movie does not indicate this clearly, rather the formation of secondary craters might be also influenced by the ocean. On the other hand, they interpreted other large pits of 50-500 m diameter located around the crater center as a different origin, devolatilization (Page 5, 13-21 and Fig. 6). Could the secondary craters also have been formed via devolatilization? Or can they exclude this and other possibilities and validate these are sure to be the secondary craters? Please also clarify the difference/similarity between these pits and their origins. We have added an additional paragraph at the end of the Discussion section to explain why these features are likely to be genetically different. See also the response to a similar point by Reviewer #1.

Additionally, I think an ejecta of 50-100 m seems a too large block (Page 8, Line 29), given that the impactor size is 160 m in diameter and the secondary crater is up to 150 m. How do the

authors estimate this block size? We acknowledge that the value of 50-100 m for individual blocks seems too large (at least at the high end of this range) – thank you for pointing this out. We have replaced this with a smaller estimate of up to 5-50 m diameter, based on scaling relationships from other studies and calculations from the Earth Impacts Effects Programme. We retain a large range for this as the impact/ejection velocity for these secondary craters is highly uncertain. Similar sized craters from the moon (see reference to Singer et al., 2020 in the revised version) are inferred to be caused by ejected blocks of around 3 m. However, this assumes an ejection speed of 2.7 km/s. The likely ejection speed in the case of Silverpit is likely much lower, at around 100 m/s at a range of 1 km from the primary crater. Adopting conventional crater scaling theory gives an estimate for blocks of up to 1/3 of the crater size, or a maximum of 50 m. We retain a large range because of the uncertainty over impact velocity for these secondary craters.

4. Formation of the concentric faults

As Stewart and Allen (2002) mentioned that the Silverpit Crater is a multi-ringed impact structure, the concentric faults on the chalk (Fig. 5) remind me of multiring basins on icy satellites (Valhalla on Callisto, Tyre and Callanish on Europa). I agree that "the rheological contrasts between different rock types" (Page 5, Line 3) could be the key. At the same time, I am wondering if this contrast could fit with the suggested mechanism of multiring basin formation (Melosh and McKinnon 1978, McKinnon and Melosh 1980); i.e., can the chalk over the mudstone correspond to the lithosphere over the asthenosphere? Are there any features shown in the impact model? Please comment on this. We agree that the unique terrestrial structure of Silverpit could have important implications for other extraterrestrial multi-ring impact structures, particularly for those on icy satellites (e.g. Tyre, Callanish on Europa), although we note that these may be genetically different to multiring basins on rocky planets (e.g. Valhalla) described primarily by Melosh and McKinnon (1978) and McKinnon and Melosh (1980). We note also that, in the case of Silverpit, the rheology is more complex, in that we have relatively unconsolidated sediment over competent chalk, over mudstone. We have added a short section to discuss this in more detail in the first part of the discussion, but note that we cannot extend this discussion significantly in a short-format paper.

Here are some other minor comments.

Page 6, Line 11: Fig. S5 is missing, or just a typo of S4? Apologies, this was a typo of S4 – now corrected.

Page 9, Line 43: How about comparing with terrestrial complex craters, not extraterrestrial ones? It is not clear what this refers to. Page 9, Line 43 states "Assuming an original thickness of between 500 m (Fig. 6b) to 600 m (the thickness in 43/25-1; Fig. S2)...". However, we have removed the reference exclusively to Earth on the first line of this paragraph, as this statement about complex craters is valid for both terrestrial and extraterrestrial craters.

Page 12, Lines 4-5: Please state the mesh size. It should be fine enough to resolve the layer and the diameter of 160 m (Line 20). Relating to this, adding the impactor's properties in Table S2 might be better. The grid dimensions have been added in the Methods section.

References:

Collins et al. 2022: <https://doi.org/10.1038/s41467-020-15269-x>
Gault and Wedekind 1978: <https://ui.adsabs.harvard.edu/abs/1978LPSC....9.3843G/abstract>
Luo et al. 2022: <https://doi.org/10.1029/2022JE007333>
McKinnon and Melosh 1980: [https://doi.org/10.1016/0019-1035\(80\)90037-8](https://doi.org/10.1016/0019-1035(80)90037-8)
Melosh and McKinnon 1978: <https://doi.org/10.1029/GL005i011p00985>

Reviewer #4 (Remarks to the Author):

This is a very well written paper which clearly demonstrates that the Silverpit structure is an impact crater. The seismic data are of very high quality and provide convincing evidence for the impact origin even without the complementary shocked mineral evidence. The authors might want to consider the following comments prior to a decision on acceptance. We thank Reviewer 4 for these positive comments and have addressed their comments below.

GENERAL COMMENTS

A crater diameter of 3 km is close to the limit separating simple craters from complex craters. The presence of a central uplift in the seismic data suggests the Silverpit crater falls into the category for complex craters and that a transient crater would have been present directly after impact that would have been somewhat smaller than the present day final diameter. Some discussion on the scaling relationships between simple and complex craters and the implications of crater type on the seismic interpretation would have been useful. Thank you for this observation. We have noted that according to our new interpretation of the crater size, Silverpit is just above the simple-to-complex transition and that in this case a central uplift and substantial gravity-driven collapse are expected. We also now note in the modelling section that the transient crater is enlarged by about 30% during crater collapse (prior to resurgence).

According to the authors PSDM was applied to the seismic data. However, all vertical sections through the seismic volume are in TWT. Is this a mistake or have the seismic data been converted back from depth to time? Data were available in both time and depth domains. However, we used the time domain PSDM data for this interpretation, as the 43/25-1 well is not within the NEP PSDM survey area. This means that the well-to-seismic tie was carried out with the older Megamerge data in the time domain. The Methods section has been expanded to discuss this.

Reflector vs. reflection: These terms seem to be used interchangeably in the MS. In my view, a reflection is something which is observed on a seismic section while a reflector is something which is present in the sub-surface. You may want to consider using these definitions in the Results and Discussion sections. We agree with the reviewer and have changed 'reflector' to 'reflection' throughout when describing the seismic observations.

SPECIFIC COMMENTS

Pg 3, lines 13-15: "Although the structure was still only partially imaged, these data allowed the

crater rim to be redefined as a much smaller 3 km in diameter, with some evidence of radial faults on one side of the central uplift.” This sentence is confusing, is the crater 3 km in diameter or is it 3 km smaller in diameter than the 20 km mentioned earlier on line 7? From Figure 1 and further reading of the MS, I see it is the former, but it would be better to be more clear here. We acknowledge that this was not sufficiently clear in the submitted version. In fact, the reported crater size in previous publications is larger than 3 km. This has been clarified, and expanded slightly to emphasise that the authors (Stewart *et al.* 2005) still considered the crater to be Cretaceous in age, and the final crater diameter to be 8 km. See also the response to Reviewer 1 on a similar point.

Figure 2b: What does the “300 m offset” label in the figure refer to? It is explained later in the discussion, but please explain in figure caption. This has now been added to the figure caption.

Pg 3, line 29: Change “rather than a decision” to “rather than a decision”. Done

Figure 5: It would be useful to be consistent with the colors for the faults. For example; green – reverse, red – normal; for this figure and the others in the MS. Also what does the dotted line in c) represent? We have been consistent with the colours of faults on the seismic profiles, with normal faults shown in black, and reverse faults in red. However, in figure 5b, we wanted to highlight the different fault families clearly: concentric normal faults, concentric reverse faults and concave-downrange faults, the latter which have a component of strike-slip motion as well. We do not show every individual fault here in detail.

The dotted line below the red reverse faults in c has been removed to avoid ambiguity.

Pg 6, lines 29-30: “in different domains of the crystal, which might that the grain additionally shows Dauphine-type twinning” Something is missing in this sentence. Also, you could already here note that the presence of shocked quartz above the CF2 reflector will be discussed later. ‘Indicate’ has been added here. A short paragraph has been added to round off this section, also referring to the later discussion on the grains above the CF2 reflector, as proposed.

Figure 8: CF1 and CF2 appear to be incorrectly labeled in the AI panel. Yes, this has been corrected.

Pg 7, lines 1-3: It appears you are assuming the impact to be vertical, not at an angle. I also interpret this to be the case looking at Figure 10 (symmetrical). You should state this here. I note later that you write the modeling assumed a vertical impact, but better to make the point at this stage. This is already mentioned in line 2 “performed using a simplified target structure and vertical trajectory to constrain impactor parameters”

Pg 8, line 24: Change “10.f” to “10f”. Changed

Point-by-point response

We thank the editor and four reviewers for their positive review of the second version of our manuscript. The final point-by-point responses are below:

REVIEWER COMMENTS

Reviewers 1, 2 and 3 have not requested any changes.

Reviewer #4:

1. Line 480: I think "in particular to correlated the surfaces with the 43/25-1 well" should be "in particular when correlating the surfaces with the 43/25-1 well" Yes, this was not clear and has been modified accordingly.

2. Line 481: I think the authors should add a sentence stating what velocity field was used to convert the PSDM image to a PSTM image. I assume it was the velocity field used for PSDM. This has now been added.